# Advancing Expert Specialization for Better MoE

**Hongcan Guo**[1][*]   **Haolang Lu**[1][*]   **Guoshun Nan**[1][†]   **Bolun Chu**[1]   **Jialin Zhuang**[1]

**Yuan Yang**[1]   **Wenhao Che**[1]   **Xinye Cao**[1]   **Sicong Leng**[2]   **Qimei Cui**[1]   **Xudong Jiang**[2]

[1]Beijing University of Posts and Telecommunications, China
[2]Nanyang Technological University, Singapore

`{ai.guohc,lhl_2507,nanguo2021}@bupt.edu.cn`

## Abstract

Mixture-of-Experts (MoE) models enable efficient scaling of large language models (LLMs) by activating only a subset of experts per input. However, we observe that the commonly used auxiliary load balancing loss often leads to expert overlap and overly uniform routing, which hinders expert specialization and degrades overall performance during post-training. To address this, we propose a simple yet effective solution that introduces two complementary objectives: (1) an orthogonality loss to encourage experts to process distinct types of tokens, and (2) a variance loss to encourage more discriminative routing decisions. Gradient-level analysis demonstrates that these objectives are compatible with the existing auxiliary loss and contribute to optimizing the training process. Experimental results over various model architectures and across multiple benchmarks show that our method significantly enhances expert specialization. Notably, our method improves classic MoE baselines with auxiliary loss by up to 23.79%, while also maintaining load balancing in downstream tasks, without any architectural modifications or additional components. Our code is available at this link.

## 1 Introduction

Large language models (LLMs) [67, 65, 62, 6] have demonstrated remarkable generalization capabilities [52, 69, 74, 73] across a wide range of tasks [53, 24], but their inference cost [15, 57] grows rapidly with scale, hindering practical deployment and efficiency. Mixture-of-Experts (MoE) [9, 3, 37] architectures alleviate this problem by activating only a subset of experts per input [19], thus enabling greater model capacity without a commensurate increase in computational overhead [22, 49, 33]. To maximize parameter utilization, MoE systems typically introduce load balancing [56, 20] objectives that encourage a more uniform routing of tokens across experts during pre-training.

While load balancing is effective in avoiding idle experts during large-scale pre-training, it often hinders model adaptation in the **post-training stage** for downstream tasks, where data distributions are narrower and more domain-specific. In such settings, token occurrences are typically concentrated within particular subspaces (e.g., numeric or symbolic tokens in math tasks), intensifying the tension between balanced routing and expert specialization. A widely observed phenomenon is that *load balancing encourages uniform expert routing across inputs*, resulting in highly overlapping token distributions [14, 79]. This overlap leads to convergence in expert representations [46], ultimately compromising the development of specialized functionalities. The lack of specialization [14] becomes particularly problematic during fine-tuning [17, 60, 2, 80] on downstream tasks with strong domain preferences, where the model struggles to adapt and exhibits degraded performance [34].

---

[*]Equal contribution.
[†]Corresponding author.

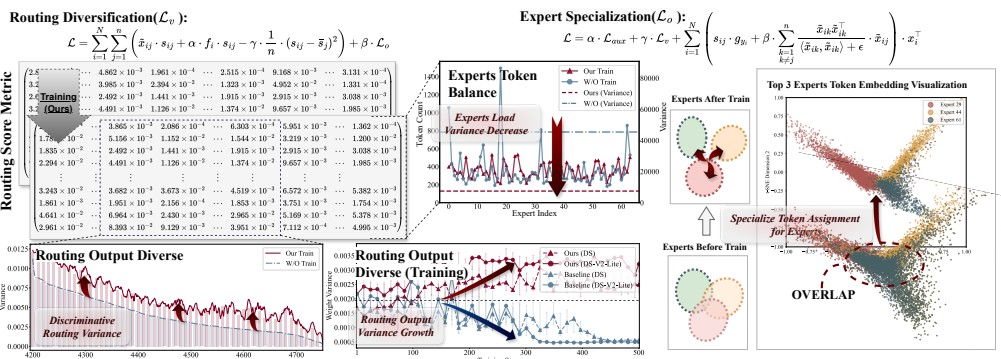

Figure 1: **Two core effects of our method. Left — Routing Diversification:** *Left-Bottom:* after training, scores show higher discrimination than the untrained model. *Right-Top:* expert load variance decrease after training. *Right-Bottom:* when training, variance increases markedly, yielding more decisive token-to-expert assignments. **Right — Expert Specialization:** *Cluster Separation:* clearer per-expert token clusters emerge after training, evidencing specialization. *Overlap:* baseline exhibits heavy token-assignment overlap across experts, which our method substantially reduces.

This highlights a core challenge in MoE post-training: the inherent conflict between *encouraging expert specialization* [50, 38, 36]and *enforcing routing uniformity* [83] via auxiliary losses. From the **expert** perspective, load-balanced routing causes overlapping training intentions across experts [14, 45, 46, 7], suppressing the development of distinct expert behaviors. From the **router** perspective, as experts become less specialized, the router receives less variation across experts, leading to increasingly uniform and less informed token-to-expert assignments [82]. These dynamics form a self-reinforcing loop: diminished specialization and uniform routing exacerbate each other over time, progressively degrading both expert expressiveness and routing quality [20]. This compounding effect reveals a deeper limitation of existing training objectives, which lack mechanisms to decouple expert specialization from the uniformity constraints imposed by auxiliary losses.

To address this challenge, we propose a gradient-based multi-objective optimization framework that promotes expert specialization and routing diversification, while preserving load balance from auxiliary loss. We introduce two complementary objectives, as shown in Figure 1: 1) **Expert Specialization**, which fosters distinct expert representations by ensuring that each expert specializes in processing different tokens. 2) **Routing Diversification**, which drives differentiated routing decisions, enabling more precise token-to-expert assignments by enhancing the variance in routing. By jointly optimizing these objectives, our method mitigates the trade-off between model performance and routing efficiency in MoE training. We demonstrate that our approach successfully achieves:

- **Enhanced expert–routing synergy**. Our joint objectives reduce expert overlap by up to 45% and increase routing score variance by over 150%, leading to clearer specialization and more discriminative expert assignment.

- **Stable load balancing**. Despite introducing new objectives, our method matches the baseline's MaxVioglobal across all models, with RMSE under 8.63 in each case.

- **Improved downstream performance**. We achieve 23.79% relative gains across 11 benchmarks and outperform all baselines on 92.42% of tasks ,all without modifying the MoE architecture.

## 2 Motivation

### 2.1 Preliminaries of MoE

In a typical MoE layer, let there be $n$ experts, and a sequence of input tokens represented by $X = \{x_1, x_2, \cdots, x_N\}$, where $N$ is the total number of tokens in the sequence. The routing score matrix after applying the top-k mechanism is denoted as:

$$\mathcal{S} = \begin{pmatrix} s_{11} & s_{12} & \cdots & s_{1n} \\ s_{21} & s_{22} & \cdots & s_{2n} \\ \vdots & \vdots & \ddots & \vdots \\ s_{N1} & s_{N2} & \cdots & s_{Nn} \end{pmatrix}, \quad \sum_{j=1}^{n} s_{ij} = 1, \quad i = 1, 2, \cdots, N \qquad (1)$$

where $s_{ij}$ represents the routing weight assigned to the $i$-th token for the $j$-th expert.

Let $F = \{f_1, f_2, \cdots, f_n\}$ represent the proportion of tokens assigned to each expert, where $f_j$ is the number of tokens assigned to the $j$-th expert. For any given MoE layer, the total loss function $\mathcal{L}$ consists of two parts, the main loss $\mathcal{L}_h$ and the auxiliary loss $\mathcal{L}_{aux}$:

$$\mathcal{L} = \mathcal{L}_h + \alpha \cdot \mathcal{L}_{aux} = \mathcal{L}_h + \alpha \sum_{j=1}^{n} f_j \cdot p_j, p_j = \sum_{i=1}^{N} s_{ij}, \tag{2}$$

where $\mathcal{L}_h$ is the loss computed from the output of the MoE layer, and $\mathcal{L}_{aux}$ is the auxiliary loss term, $\alpha$ denotes the weighting coefficient for the auxiliary loss. Here, $p_j$ represents the total routing score for the $j$-th expert, which is the sum of the routing weights for all tokens assigned to that expert.

## 2.2 Observations

***Obs I (Expert Overlap):** Introduction of the auxiliary loss function leads to a more homogenized distribution of tokens across experts, which may reduce the distinctiveness of each expert.*

It has been observed that the auxiliary loss function is independent of the expert parameter matrices $\theta_{E_j}$. Therefore, for the $j$-th expert, its gradient can be written as:

$$\frac{\partial \mathcal{L}}{\partial \theta_{E_j}} = \frac{\partial \mathcal{L}_h}{\partial \theta_{E_j}} + \alpha \cdot \frac{\partial \mathcal{L}_{aux}}{\partial \theta_{E_j}} = \frac{\partial \mathcal{L}}{\partial y_h} \cdot \frac{\partial y_h}{\partial \theta_{E_j}} = \sum_{i=1}^{N} x_i \cdot s_{ij}, j = 1, 2, \cdots, n. \tag{3}$$

where $\theta_{E_j}$ is the parameter matrix of the $j$-th expert, and $y_h$ is the output of the MoE layer. During gradient descent, the addition of the auxiliary loss $\mathcal{L}_{aux}$ forces the routing mechanism to evenly distribute the tokens across experts as much as possible.

This results in input token $x_i$ being assigned to an expert that may not be semantically aligned with it, causing an unintended gradient flow to expert $j$. Mathematically, after applying the top-k mechanism, the routing score $s_{ij}$ transitions from 0 to a non-zero value, introducing gradients from tokens that originally had no affinity with expert $j$.

***Obs II (Routing Uniformity):** As training progresses, the routing output tends to become more uniform, with the expert weight distribution gradually converging towards an equal allocation.*

To understand this phenomenon, we first examine the source of gradients with respect to the routing parameters $\theta_R$. Since the routing mechanism produces only the score matrix $\mathcal{S} = s_{ij}$, the gradient $\partial \mathcal{L} / \partial \theta_R$ can be written as:

$$\frac{\partial L}{\partial \theta_R} = \frac{\partial \mathcal{L}_h}{\partial \theta_R} + \alpha \cdot \frac{\partial \mathcal{L}_{aux}}{\partial \theta_R} = \sum_{i=1}^{N} x_i \sum_{j=1}^{n} \theta_{E_j} \cdot \frac{\partial s_{ij}}{\partial \theta_R} + \alpha \cdot \sum_{j=1}^{n} f_j \sum_{i=1}^{N} \frac{\partial s_{ij}}{\partial \theta_R}, \tag{4}$$

where $x_i \cdot \theta_{E_j}$ represents the output of expert $j$ for token $x_i$, and $f_j$ denotes the frequency with which expert $j$ is selected. This formulation reveals that the routing gradient is primarily influenced by the expert outputs and the token distribution across experts.

The auxiliary loss $\mathcal{L}_{aux}$ is introduced to encourage balanced token assignment by optimizing the uniformity of $f_j$. However, since $f_j$ is non-differentiable, direct optimization is not feasible. Instead, a surrogate variable $p_j$, which is differentiable and positively correlated with $f_j$, is employed to approximate the objective and enable gradient flow back to the routing network.

As training proceeds, the optimization objective increasingly favors the uniformity of $p_j$, which drives $f_j$ toward an even distribution. Moreover, as discussed in Observation I, incorrect token assignments caused by auxiliary regularization introduce overlapping gradients among experts, increasing the similarity of $x_i \cdot \theta_{E_j}$ across different $j$.

***Obs III (Expert–Routing Interaction):** While **Obs I** concerns expert specialization, while **Obs II** reflects the uniformity of routing. These two effects interact during training, jointly driving the model toward degraded performance.*

- *Expert-side interference caused by Obs I leads to blurred specialization.* Tokens are assigned to mismatched experts, and the resulting gradient interference reduces expert distinctiveness. As the

routing weights become more uniform, different experts receive similar gradients from the same tokens, increasing their functional overlap.

- *This expert similarity feeds back into the routing mechanism.* As expert outputs become less distinguishable, the routing network finds fewer cues to differentiate among experts, leading to even more uniform weight distributions. This promotes random top-$k$ selection and further misalignment between tokens and their optimal experts.

Together, this loop gradually steers the model toward more uniform token allocation and reduced expert specialization, highlighting potential opportunities for improving the routing strategy and expert assignment.

## 3 Method

Based on the observations above, we propose the following design to mitigate **expert overlap** and **routing uniformity**, the overall loss function $\mathcal{L}$ is defined as follows:

$$\mathcal{L} = \mathcal{L}_h + \mathcal{L}_{balance}, \quad \mathcal{L}_{balance} = \alpha \cdot \mathcal{L}_{aux} + \beta \cdot \mathcal{L}_o + \gamma \cdot \mathcal{L}_v, \tag{5}$$

where $\mathcal{L}_{aux}$ represents the existing auxiliary loss, with coefficient $\alpha$, and the newly introduced orthogonality loss $\mathcal{L}_o$ and variance loss $\mathcal{L}_v$ (see Subsec 3.1), with coefficients $\beta$ and $\gamma$ respectively. It is worth noting that the theoretical complementarity of these optimization objectives, rather than any inherent conflict, is formally analyzed and demonstrated in Subsection 3.2.

### 3.1 Implementations of Losses $\mathcal{L}_o$ and $\mathcal{L}_v$

In this section, we introduce two critical loss functions $\mathcal{L}_o$ and $\mathcal{L}_v$ that act on the expert and router components, respectively.

**Expert Specialization.** We introduce an orthogonalization objective that encourages independent expert representations. Specifically, we design the following orthogonality loss:

$$\mathcal{L}_o = \sum_{i=1}^{N} \sum_{j=1}^{n} \sum_{\substack{k=1 \\ k \neq j}}^{n} \left\| \frac{\langle \tilde{x}_{ij}, \tilde{x}_{ik} \rangle}{\langle \tilde{x}_{ik}, \tilde{x}_{ik} \rangle + \epsilon} \tilde{x}_{ik} \right\|^2, \quad \tilde{x}_{ij} = x_i \cdot \theta_{E_j} \cdot \mathbb{I}_{\{s_{ij} > 0\}}, \tag{6}$$

where $\langle \cdot \rangle$ denotes the inner product between two vectors, and $\mathbb{I}{s_{ij}} > 0$ is an indicator function that evaluates to 1 when $s_{ij} > 0$ and 0 otherwise. Here, $\tilde{x}_{ij}$ represents the output of expert $j$ for token $x_i$ after the top-$k$ routing selection.

The orthogonality loss $\mathcal{L}_o$ reduces the overlap between different expert outputs within the same top-$k$ group by minimizing their projections onto each other. This encourages experts to develop more distinct representations, promoting specialization in processing different token types.

**Routing Diversification.** We introduce a variance-based loss to encourage more diverse routing decisions and promote expert specialization. Specifically, we define the variance loss as:

$$\mathcal{L}_v = -\sum_{i=1}^{N} \sum_{j=1}^{n} \frac{1}{n} \cdot (s_{ij} - \bar{s}_j)^2, \bar{s}_j = \frac{1}{N} \cdot \sum_{i=1}^{N} s_{ij}, \tag{7}$$

where $\bar{s}_j$ denotes the average routing score for expert $j$ across the batch. By maximizing the variance of routing scores, $\mathcal{L}_v$ discourages uniform token-to-expert assignments and encourages more deterministic and distinct routing patterns, thereby facilitating expert specialization.

### 3.2 Compatibility of Multi-Objective Optimization

In this section, we analyze how each component influences the optimization dynamics of expert parameters $\theta_{E_j}$ and routing parameters $\theta_R$ during training. Meanwhile, we will focus on the optimization and compatibility of the two losses $L_o$ and $L_v$ with respect to load balancing and expert specificity. The following two key questions guide our analysis.

***Balancing Expert and Routing.*** *How can expert ($\mathcal{L}_o$) and routing ($\mathcal{L}_v$) optimizations be designed to complement each other without compromising their respective objectives?*

We first demonstrate that $\mathcal{L}_o$ and $\mathcal{L}_v$ are compatible in their optimization directions within MoE, then show that they mutually reinforce each other.

**Mutually Compatible.** We elaborate on the compatibility of $\mathcal{L}_o$ and $\mathcal{L}_v$ from the perspectives of expert and Routing.

From the **expert perspective**, we observe that the auxiliary loss $\mathcal{L}_{aux}$ and the variance loss $\mathcal{L}_v$ do not directly contribute gradients to the expert parameter matrix $\theta_{E_j}$. Consequently, the gradient of the total loss with respect to $\theta_{E_j}$ is derived solely from the primary task loss $\mathcal{L}_h$ and the orthogonality loss $\mathcal{L}_o$:

$$\frac{\partial \mathcal{L}}{\partial \theta_{E_j}} = \sum_{i=1}^{N} \left( s_{ij} \cdot g_{y_i} + \beta \cdot \sum_{\substack{k=1 \\ k \neq j}}^{n} \frac{\tilde{x}_{ik}\tilde{x}_{ik}^{\top}}{\langle \tilde{x}_{ik}, \tilde{x}_{ik} \rangle + \epsilon} \cdot \tilde{x}_{ij} \right) \cdot x_i^{\top} \tag{8}$$

Here, $g_{y_i} = \nabla_{y_i}\mathcal{L}_h$ denotes the gradient of the primary task loss with respect to the model output. This gradient is influenced by both the routing score $s_{ij}$ and the expert representation $\tilde{x}_{ij}$. As training progress, the variance of expert weights increases, and the gradient encourages stronger preferences in different directions for each token.

From the **routing perspective**, we notice that $\mathcal{L}_o$ does not affect the gradient with respect to routing parameters $\theta_R$. The gradient of the total loss with respect to $\theta_R$ is:

$$\frac{\partial \mathcal{L}}{\partial \theta_R} = \frac{\partial \mathcal{L}}{\partial s_{ij}} \cdot \frac{\partial s_{ij}}{\partial \theta_R} = \sum_{i=1}^{N} \sum_{j=1}^{n} \left( \tilde{x}_{ij} + \alpha \cdot f_j - \gamma \cdot \frac{2(N-1)}{nN} \cdot (s_{ij} - \bar{s}_j) \right) \cdot \frac{\partial s_{ij}}{\partial \theta_R}. \tag{9}$$

This gradient is influenced by expert representations $\tilde{x}_{ij}$, expert load $f_j$, and routing weights $s_{ij}$. As the model converges, the expert load $f_j$ becomes more balanced, and the variance of routing weights $s_{ij}$ increases. Orthogonalizing expert representations causes the routing gradients to flow in more orthogonal directions, making the weight allocation more biased towards the representations and increasing the weight variance.

> **Summary.** Expert parameters $\theta_{E_j}$ are solely influenced by the gradients of $\mathcal{L}_o$ without conflict. While routing parameters $\theta_R$ are affected by both $\mathcal{L}_o$ and $\mathcal{L}_v$, the objectives of these two losses (orthogonality-friendliness vs. score diversification) remain non-conflicting.

**Mutually Reinforcing.** $\mathcal{L}_o$ aims to encourage the effective output vectors of different selected experts $j$ and $k$ to tend to be orthogonal for the same input token $x_i$, i.e., $\langle \tilde{x}_{ij}, \tilde{x}_{ik} \rangle \approx 0$. The learning signal for the routing mechanism partially originates from the gradient of the primary task loss $\mathcal{L}_h$ with respect to the routing score $s_{ij}$:

$$\frac{\partial \mathcal{L}}{\partial s_{ij}} = \underbrace{g_{y_i}^T \tilde{x}_{ij}}_{\text{from } \mathcal{L}_h} + \underbrace{\alpha \frac{\partial \mathcal{L}_{\text{aux}}}{\partial s_{ij}}}_{\text{from } \mathcal{L}_{\text{aux}}} - \underbrace{\gamma \frac{2(N-1)}{nN}(s_{ij} - \bar{s}_j)}_{\text{from } \mathcal{L}_v}, \quad y_i = \sum_j s_{ij}\tilde{x}_{ij}, \quad g_{y_i} = \frac{\partial \mathcal{L}_h}{\partial y_i} \tag{10}$$

Assuming $p_{ij} = g_{y_i}^T \tilde{x}_{ij}$, when the expert outputs tend to be orthogonal, for any given task gradient $g_{y_i}$, the projections $p_{ij}$ onto these approximately orthogonal expert outputs are more likely to exhibit significant differences. The increased variance of the primary task-related signals $p_{ij}$ implies that the routing mechanism receives more discriminative and stronger learning signals, which creates more favorable conditions for $\mathcal{L}_v$ to achieve diversification of routing scores.

$\mathcal{L}_v$ enhances the diversity of routing scores $s_{ij}$ by optimizing routing parameters $\theta_R$. Meanwhile, due to the influence of $\mathcal{L}_o$'s gradient $\beta \frac{\partial \mathcal{L}_o}{\partial s_{ij}}$ on $\theta_R$, routing tends to assign more specialized token subsets $T_j$ to each expert $j$. Expert parameters $\theta_{E_j}$ learn the unique features of tokens within $T_j$, leading to gradual functional divergence among experts, thereby promoting expert orthogonality.

> **Summary.** $\mathcal{L}_o$ induces orthogonal expert outputs $\tilde{x}_{ij}$, enhances the discriminative power of routing signals $g_{y_i}^T \tilde{x}_{ij}$, and generates diverse routing scores $s_{ij}$ to support $\mathcal{L}_v$. Meanwhile, $\mathcal{L}_v$ drives experts to specialize in distinct token subsets via $s_{ij}$ and promotes parameter divergence of $\theta_{E_j}$ to support $\mathcal{L}_o$. Together, they form a mutually reinforcing cycle.

***Multi-Objective Optimization.*** *How do expert and routing maintain their balance while enhancing* $\mathcal{L}_{aux}$ *and* $\mathcal{L}_h$ *independently, ensuring mutually beneficial performance improvements?*

**Lemma 1** *Let* $\mathcal{S} \in \mathcal{R}^{N \times n}$ *be a matrix that satisfies following conditions: each row sums to 1, each row contains* $k$ *non-zero elements and* $n - k$ *zero elements. Then, there always exists a state in which the following two objectives are simultaneously optimized: 1. The sum of the elements in each column tends to the average value* $\frac{N}{n}$; *2. The variance of the non-zero elements in each row increases.*

**Lemma 2** *For two sets of points* $\mathcal{A}$ *and* $\mathcal{B}$ *of equal size, it is always possible to partition* $\mathcal{A} \cup \mathcal{B}$ *such that* $\mathcal{A} \cap \mathcal{B} = \varnothing$ *and* $|\mathcal{A}| = |\mathcal{B}|$.

The overall objective function $\mathcal{L}$ optimizes four key dimensions: accurate data fitting($\mathcal{L}_h$), expert orthogonalization($\mathcal{L}_o$), balanced expert routing weights($\mathcal{L}_{aux}$), and increased variance in routing outputs($\mathcal{L}_v$). Our core objective is to achieve an **optimal balance by jointly optimizing these multiple objectives**, ensuring they complement each other for enhanced model performance.

As shown by Lemma 1, expert load $f_j$ and routing weights $s_{ij}$ can be optimized together. As demonstrated in Lemma 2, the objectives of orthogonalization and load balancing are not in conflict and can be jointly optimized. Thus, both expert and routing modifications can be optimized alongside load balancing (balanced expert routing weights).

Moreover, orthogonalization enhances routing weight variance, in turn, improves expert specialization (as discussed in Section 2.2). This leads to more distinctive expert representations, aligning with performance (accurate data fitting) improvements when optimized together.

# 4 Experiments

In this section, we conduct experiments to address the following research questions:

- **RQ1**: Does introducing the orthogonality loss ($\mathcal{L}_o$) and variance loss ($\mathcal{L}_v$) lead to better overall performance in downstream tasks compared to baseline approaches?

- **RQ2**: To what extent does our method maintain expert load balancing during training?

- **RQ3**: How do the orthogonality loss ($\mathcal{L}_o$) and variance loss ($\mathcal{L}_v$) interact with each other, and what are their respective and joint impacts on expert specialization and routing behavior?

- **RQ4**: What are the individual and combined contributions of $\mathcal{L}_o$, $\mathcal{L}_v$, and the auxiliary loss $\mathcal{L}_{aux}$ to the final model performance?

## 4.1 Experimental Setup

**Environment.** All experiments are performed on a CentOS Linux 7 server with PyTorch 2.3. The hardware specifications consist of 240GB of RAM, a 16-core Intel Xeon CPU, and two NVIDIA A800 GPUs, each having 80GB of memory. Implementation details are provided in the Appendix F.

**Datasets.** We evaluate our method on a total of **11 benchmarks**. Specifically, we use the training sets from Numina [41], GLUE [66], and the FLAN collection [72] to train our models. Our benchmarks include: ❶ **Mathematics**: GSM8K [12], MATH500 [44], and Numina [41]; ❷ **Multi-Domain Tasks**: MMLU [31, 30], MMLU-pro [70], BBH [63], GLUE [66]; LiveBench [76] and GPQA [59]. ❸ **Code generation**: HumanEval [10] and MBPP [4]. We group training and test sets by language, reasoning, science, math, and code to match downstream evaluation needs. Detail in Appendix D.

**Baselines.** We compare our method with **4 existing MoE training strategies**. With Aux Loss [46] applies auxiliary load-balancing losses during routing to encourage expert utilization diversity. GShard [39] introduces a foundational sparse expert framework with automatic sharding and routing; ST-MoE [85] enhances training stability via router dropout and auxiliary losses; Loss-Free Balancing [68] achieves balanced expert routing without auxiliary objectives. Detail in Appendix G.

**Metrics.** We employ **6 evaluation metrics** to test our method in terms of accuracy, expert load balancing (MaxVio$_{global}$ [68]), clustering quality (Silhouette Coefficient), expert specialization (Expert Overlap), routing stability (Routing Variance), and prediction error (RMSE). Detail in Appendix E.

Table 1: **Performance on different downstream tasks.** The table shows accuracies of methods across models and downstream tasks. Notably, **we categorize sub-downstream tasks in Multi-Domain and ensure training/evaluation sets are domain-aligned**, following downstream task requirements.

| Method | Model | Multi-Domain (Avg.) | | | | | | Code | | Math | | |
|---|---|---|---|---|---|---|---|---|---|---|---|---|
| | | MMLU | MMLU-pro | BBH | GLUE | Livebench | GPQA | HumanEval | MBPP | GSM8K | MATH500 | NuminaTest |
| With Aux Loss | DeepSeek-MoE-16B | $29.27_{\pm0.10}$ | $19.47_{\pm2.50}$ | $26.92_{\pm2.30}$ | $49.26_{\pm0.40}$ | $7.43_{\pm0.10}$ | $21.15_{\pm0.40}$ | $51.52_{\pm1.50}$ | $31.36_{\pm1.10}$ | $15.70_{\pm2.40}$ | $5.47_{\pm1.50}$ | $14.99_{\pm2.40}$ |
| Loss-Free Balancing | | $30.71_{\pm2.10}$ | $16.81_{\pm0.70}$ | $32.99_{\pm1.00}$ | $49.60_{\pm1.30}$ | $\underline{9.79}_{\pm0.20}$ | $20.63_{\pm1.60}$ | $53.16_{\pm2.40}$ | $32.80_{\pm1.40}$ | $21.28_{\pm0.40}$ | $5.83_{\pm1.30}$ | $\underline{17.23}_{\pm1.60}$ |
| GShard | | $27.05_{\pm2.00}$ | $\underline{20.48}_{\pm0.60}$ | $29.83_{\pm1.80}$ | $53.83_{\pm0.70}$ | $8.69_{\pm1.20}$ | $\underline{24.28}_{\pm2.30}$ | $\underline{57.75}_{\pm2.20}$ | $34.50_{\pm1.70}$ | $27.12_{\pm1.30}$ | $\underline{8.20}_{\pm1.50}$ | $16.99_{\pm0.70}$ |
| ST-MOE | | $\mathbf{34.23}_{\pm2.20}$ | $19.71_{\pm0.80}$ | $\underline{36.91}_{\pm1.90}$ | $\underline{54.56}_{\pm2.30}$ | $6.48_{\pm0.70}$ | $20.35_{\pm0.90}$ | $53.28_{\pm1.60}$ | $\underline{36.34}_{\pm1.50}$ | $\underline{30.10}_{\pm2.00}$ | $7.08_{\pm0.40}$ | $15.48_{\pm1.20}$ |
| **Ours** | | $\underline{33.35}_{\pm2.20}$ | $\mathbf{24.87}_{\pm1.20}$ | $\mathbf{37.52}_{\pm1.40}$ | $\mathbf{60.01}_{\pm1.00}$ | $\mathbf{11.00}_{\pm1.70}$ | $\mathbf{25.15}_{\pm0.40}$ | $\mathbf{63.30}_{\pm0.70}$ | $\mathbf{40.03}_{\pm0.40}$ | $\mathbf{35.00}_{\pm1.00}$ | $\mathbf{10.82}_{\pm0.30}$ | $\mathbf{20.41}_{\pm0.10}$ |
| With Aux Loss | DeepSeek-V2-Lite | $\underline{33.23}_{\pm2.10}$ | $28.40_{\pm0.20}$ | $34.80_{\pm1.40}$ | $35.97_{\pm0.20}$ | $11.70_{\pm0.50}$ | $24.92_{\pm0.80}$ | $40.24_{\pm0.80}$ | $\underline{41.23}_{\pm0.20}$ | $44.79_{\pm2.10}$ | $42.03_{\pm1.40}$ | $42.01_{\pm1.90}$ |
| Loss-Free Balancing | | $30.23_{\pm0.80}$ | $\underline{30.75}_{\pm2.10}$ | $34.21_{\pm1.10}$ | $\underline{39.83}_{\pm1.80}$ | $10.15_{\pm1.10}$ | $\underline{26.33}_{\pm0.60}$ | $41.28_{\pm1.40}$ | $36.02_{\pm2.30}$ | $43.35_{\pm0.70}$ | $39.76_{\pm1.10}$ | $43.90_{\pm1.10}$ |
| GShard | | $30.86_{\pm1.10}$ | $29.13_{\pm0.80}$ | $37.67_{\pm0.30}$ | $38.89_{\pm1.00}$ | $\underline{13.17}_{\pm1.80}$ | $24.34_{\pm2.10}$ | $\mathbf{45.36}_{\pm1.60}$ | $37.00_{\pm2.10}$ | $45.39_{\pm1.50}$ | $43.61_{\pm2.10}$ | $43.25_{\pm0.70}$ |
| ST-MOE | | $32.68_{\pm2.10}$ | $30.28_{\pm2.10}$ | $\underline{38.78}_{\pm0.90}$ | $38.27_{\pm1.00}$ | $10.60_{\pm2.30}$ | $22.33_{\pm0.40}$ | $44.10_{\pm0.20}$ | $39.72_{\pm2.30}$ | $\underline{47.78}_{\pm1.80}$ | $\underline{46.74}_{\pm0.50}$ | $\underline{48.65}_{\pm0.70}$ |
| **Ours** | | $\mathbf{35.59}_{\pm0.50}$ | $\mathbf{37.37}_{\pm0.20}$ | $\mathbf{38.84}_{\pm1.70}$ | $\mathbf{41.20}_{\pm2.00}$ | $\mathbf{14.60}_{\pm2.50}$ | $\mathbf{28.76}_{\pm0.10}$ | $\underline{43.58}_{\pm0.30}$ | $\mathbf{43.53}_{\pm2.40}$ | $\mathbf{50.94}_{\pm2.40}$ | $\mathbf{49.33}_{\pm2.40}$ | $\mathbf{50.67}_{\pm1.10}$ |
| With Aux Loss | Moonlight-16B-A3B | $35.82_{\pm1.40}$ | $\mathbf{36.10}_{\pm1.50}$ | $47.17_{\pm0.70}$ | $26.16_{\pm1.20}$ | $15.84_{\pm1.70}$ | $30.72_{\pm1.90}$ | $63.61_{\pm1.90}$ | $47.34_{\pm1.50}$ | $82.32_{\pm1.50}$ | $57.03_{\pm1.60}$ | $45.41_{\pm0.40}$ |
| Loss-Free Balancing | | $27.40_{\pm0.10}$ | $31.91_{\pm2.10}$ | $42.45_{\pm0.50}$ | $32.97_{\pm1.60}$ | $\underline{20.05}_{\pm2.40}$ | $29.27_{\pm1.80}$ | $62.93_{\pm2.50}$ | $44.92_{\pm1.30}$ | $79.34_{\pm0.70}$ | $\underline{57.77}_{\pm0.50}$ | $42.82_{\pm0.10}$ |
| GShard | | $\underline{36.06}_{\pm0.90}$ | $30.65_{\pm0.50}$ | $\underline{49.20}_{\pm1.70}$ | $\underline{34.46}_{\pm2.40}$ | $13.97_{\pm2.30}$ | $\underline{31.13}_{\pm1.10}$ | $64.50_{\pm1.50}$ | $\mathbf{49.85}_{\pm0.50}$ | $\underline{84.62}_{\pm0.80}$ | $56.09_{\pm2.20}$ | $47.18_{\pm2.30}$ |
| ST-MOE | | $33.03_{\pm0.90}$ | $26.83_{\pm1.70}$ | $46.78_{\pm0.30}$ | $30.18_{\pm1.50}$ | $16.99_{\pm1.70}$ | $30.93_{\pm1.50}$ | $\underline{66.04}_{\pm1.60}$ | $\underline{47.97}_{\pm2.20}$ | $84.45_{\pm0.90}$ | $57.61_{\pm1.60}$ | $\underline{49.42}_{\pm2.10}$ |
| **Ours** | | $\mathbf{40.36}_{\pm2.20}$ | $\underline{34.90}_{\pm0.30}$ | $\mathbf{52.42}_{\pm1.80}$ | $\mathbf{37.01}_{\pm1.10}$ | $\mathbf{20.85}_{\pm1.10}$ | $\mathbf{32.01}_{\pm0.90}$ | $\mathbf{70.64}_{\pm0.20}$ | $47.77_{\pm1.00}$ | $\mathbf{87.62}_{\pm2.20}$ | $\mathbf{59.64}_{\pm0.20}$ | $\mathbf{52.88}_{\pm1.70}$ |

**Setup.** Each benchmark is fine-tuned separately on 6,000 high-quality examples, primarily from the official training split and supplemented when necessary. Answers are generated using strong teacher models (OpenAI o3-mini and DeepSeek R1) and manually verified for correctness. Fine-tuning is limited to three epochs (∼550 steps) to prevent overfitting.

All experiments adopt LoRA-based fine-tuning, with LoRA modules inserted into both router and expert layers to enable joint optimization. A rank of 32 is used to approximate full-model updates. Detailed configurations, including optimizer, batch size, and learning rate, are provided in Appendix H.2.

### 4.2 Performance in Downstream Tasks (RQ1)

To verify that our $\mathcal{L}_{\text{balance}}$ enhances model performance in downstream task scenarios through expert orthogonality and routing output diversification, as shown in Table 1, we design downstream task scenarios on 11 well-known benchmarks and validate our method against four baseline methods with distinct loss designs on three widely used MoE models. We make the following observations:

**Obs.❶ Baseline methods without guidance for expert specialization exhibit varied performance and fail to effectively improve downstream task performance.** As shown in Table 1, the four baseline methods show no clear overall performance ranking across the 11 tasks, with performance variations within 2% in many tasks. Their overall performance is significantly lower than our method, demonstrating no potential to improve downstream task performance.

**Obs.❷ Our method guiding expert specialization effectively enhances model performance in downstream tasks.** As shown in Table 1, we achieve state-of-the-art (SOTA) results in over 85% of the 33 tasks across the three models. In some tasks, the average across multiple measurements even outperforms the next-best method by nearly 7%. Extensive experiments indicate that our method significantly improves model performance in downstream task scenarios by enhancing expert specialization. More results on additional baselines and MoE architectures are provided in Appendix I.

### 4.3 Load Balancing (RQ2)

To verify that our newly added losses $\mathcal{L}_v$ and $\mathcal{L}_o$ do not affect the load balancing effect, we conduct statistical measurements on the load balancing of all combinations of $\mathcal{L}_{aux}$, $\mathcal{L}_v$, and $\mathcal{L}_o$ across various models during training.

Figure 2 shows the variation of $MaxVio_{\text{global}} \downarrow$ across training steps for different loss combinations, as well as the RMSE of differences between our method and other combinations. We make the following observations:

**Obs.❸ Loss combinations without $\mathcal{L}_{\text{aux}}$ exhibit significantly worse load balancing performance than those with $\mathcal{L}_{\text{aux}}$.** As shown in Figure 2, across three distinct models, the $MaxVio_{\text{global}}$ of the w/o all method (with no losses added) is significantly higher than that of other methods, indicating

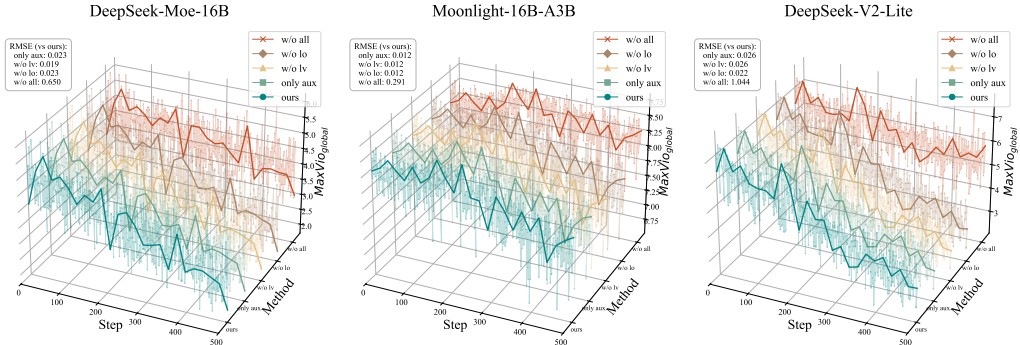

Figure 2: **Variation of Load Balancing.** The figure illustrates the variation of load balancing during training across three distinct models for different methods. Method represents the combination of $\mathcal{L}_{\text{aux}}$, $\mathcal{L}_{\text{o}}$, and $\mathcal{L}_{\text{v}}$; Step denotes the number of training steps; $MaxVio_{\text{global}} \downarrow$ serves as the metric for load balancing; and RMSE is the metric for measuring the similarity between two curves.

notably poorer load balancing. In particular, for the `DeepSeek-V2-Lite` model, the method without $\mathcal{L}_{\text{aux}}$ converges to 6.14, whereas methods with $\mathcal{L}_{\text{aux}}$ converge to 2.48, demonstrating that loss combinations containing $\mathcal{L}_{\text{aux}}$ achieve significantly better load balancing.

**Obs.❹ Incorporating any combination of $\mathcal{L}_v$ and $\mathcal{L}_o$ into $\mathcal{L}_{\text{aux}}$ does not affect load balancing.** As shown in Figure 2, for methods with $\mathcal{L}_{\text{aux}}$, the trends of "only aux" (no additional losses), "w/o lv" (only $\mathcal{L}_o$), "w/o lo" (only $\mathcal{L}_v$), and "ours" (both $\mathcal{L}_v$ and $\mathcal{L}_o$) are nearly identical. Additionally, the RMSE (root mean squared error) of our method relative to other baselines does not exceed 0.03, further corroborating the conclusion that the combination of $\mathcal{L}_v$ and $\mathcal{L}_o$ does not impact load balancing.

## 4.4 Behaviors of Experts and Routing (RQ3)

To verify that $\mathcal{L}_v$ and $\mathcal{L}_o$ can jointly promote expert orthogonality and routing score diversification, following the method setup in Section 4.3, we will conduct evaluations of expert orthogonality and measurements of routing score diversification for different loss combinations.

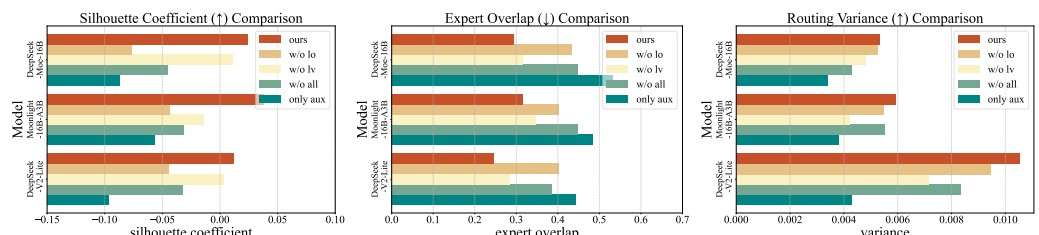

Figure 3: **Behaviors of Experts and Routing.** The figure demonstrates the behavioral states of experts and routing across different methods. The first two subplots, Silhouette Coefficient and Expert Overlap, measure the degree of expert orthogonality, while the last subplot, Routing Variance, evaluates the diversity of routing outputs.

As shown in Figure 3, the first two subplots demonstrate the orthogonality of experts, while the last subplot illustrates the diversification of routing outputs. We make the following observations:

**Obs.❺ $\mathcal{L}_o$ directly promotes expert orthogonality, and $\mathcal{L}_v$ also aids in expert orthogonality.** As shown in the first two panels of Figure 3, our method with both $\mathcal{L}_o$ and $\mathcal{L}_v$ achieves state-of-the-art (SOTA) results across three models, with Expert Overlap even dropping below 0.3. The method with only $\mathcal{L}_o$ and $\mathcal{L}_{\text{aux}}$ (w/o lv) consistently ranks second-best, indicating that $\mathcal{L}_o$ has a more significant impact on expert orthogonality. Notably, the method with only $\mathcal{L}_v$ and $\mathcal{L}_{\text{aux}}$ (w/o lo) significantly outperforms the method with only $\mathcal{L}_{\text{aux}}$ across all three models, confirming that $\mathcal{L}_v$ also contributes to expert orthogonality.

**Obs.❻** $\mathcal{L}_v$ **directly enhances routing output diversification, and** $\mathcal{L}_o$ **also supports this diversification.** Similarly, our method exhibits the highest routing score variance (exceeding 0.010), followed by the method with only $\mathcal{L}_v$ and $\mathcal{L}_{aux}$, while the method with only $\mathcal{L}_{aux}$ performs worst. This strongly supports the conclusion.

**Obs.❼** $\mathcal{L}_{aux}$ **leads to higher expert overlap and more homogeneous routing outputs.** Compared to the w/o all method (no losses), the aux only method (with only $\mathcal{L}_{aux}$) shows a Silhouette Coefficient that is over 0.05 higher and a routing output variance that is 0.0045 higher. This indicates that w/o all exhibits significantly greater expert orthogonality and routing output diversification than aux only.

## 4.5 Ablation among Losses (RQ4)

To demonstrate that both $\mathcal{L}_o$ and $\mathcal{L}_v$ have positive effects on the model's performance in downstream task scenarios, and their combination synergistically enhances each other's efficacy, we design ablation experiments for these two losses on three models.

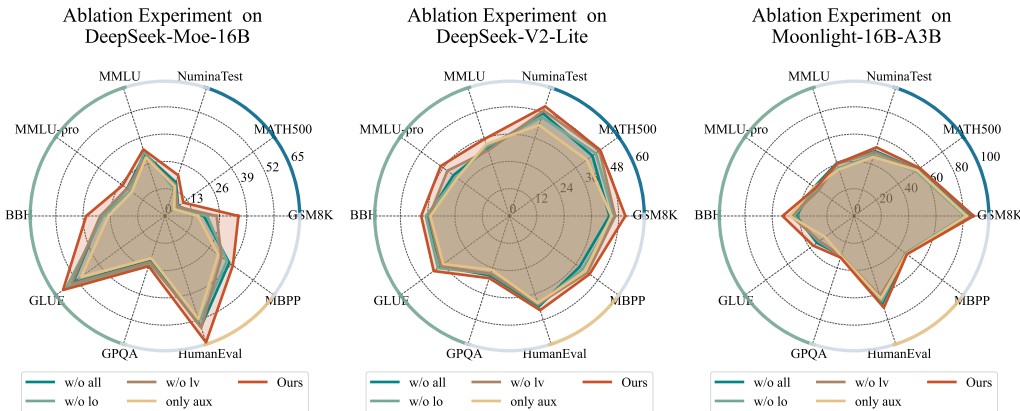

Figure 4: **Ablation Experiments.** The figure illustrates the performance differences of different ablation method combinations across three models on various benchmarks. The vertices on the circles represent the corresponding benchmark names, with the same type connected by the same color. The numbers inside the circles denote the accuracy represented by each circle.

Figure 4 illustrates the performance of different ablation method combinations across various downstream tasks. We make the following observations:

**Obs.❽ The combination of** $\mathcal{L}_o$ **and** $\mathcal{L}_v$ **significantly enhances model performance in downstream tasks, and each loss individually also improves performance.** Our method (combining $\mathcal{L}_o$ and $\mathcal{L}_v$) exhibits the largest coverage area across all three models, nearly encompassing other methods. When either $\mathcal{L}_o$ or $\mathcal{L}_v$ is ablated (i.e., w/o lv or w/o lo), the coverage areas of these methods are larger than that of the only aux method (with only $\mathcal{L}_{aux}$), indicating performance improvements over the baseline.

**Obs.❾** $\mathcal{L}_{aux}$ **impacts model performance on downstream tasks.** Figure 4 clearly shows that the only aux method (with only $\mathcal{L}_{aux}$) is nearly entirely enclosed by other methods across all three models, consistently exhibiting the smallest coverage area. Notably, the w/o all method (with no losses) achieves performance improvements and a larger coverage area than the only aux method when $\mathcal{L}_{aux}$ is removed, supporting this conclusion.

Beyond the ablation results in Fig. 4, we further conduct a sensitivity analysis on the loss-weight coefficients $\alpha$, $\beta$, and $\gamma$. The detailed results and discussions are provided in Appendix H.1.

# 5 Related Work

**Auxiliary Losses in MoE Training.** Auxiliary losses [39, 85] are commonly used to prevent expert collapse by encouraging balanced expert utilization [14]. Early approaches focus on suppressing routing imbalance, while later works [81] introduce capacity constraints or multi-level objectives to separate routing stability from load balancing [65, 39, 20]. Recent methods [75] further reduce

manual tuning by dynamically adjusting auxiliary weights or replacing them with entropy-based routing [42]. However, fixed-rule strategies may underutilize expert capacity, and dynamic schemes can introduce instability or overhead, making robust balancing still a challenge [32, 68].

**Orthogonality in MoE.** Orthogonalization [47, 28] improves expert diversity by encouraging independent representations [29]. Some methods [54, 84, 51] regularize expert weights directly, while others [14, 29] assign experts to disentangled subspaces based on task semantics. Recent routing-based approaches [47, 58] also impose orthogonality on token-to-expert assignments to reduce redundancy. Nonetheless, static constraints [11] often fail to adapt to dynamic inputs, and dynamic ones [78, 35, 25, 64] may conflict with balancing, complicating expert allocation [32, 82, 27, 68]. Our work addresses these tensions by integrating orthogonalization and balance into a unified, gradient-consistent optimization framework.

## 6 Limitation & Future Discussion

While $\mathcal{L}_{balance}$ balances load and enhances performance in downstream tasks, its potential in other domains remains unexplored. Specifically, it could be extended to visual models, as suggested in recent work [26], and multimodal or full-modal settings [8], offering opportunities for cross-domain applications. Additionally, investigating $\mathcal{L}_{balance}$ within lightweight MoE fine-tuning, such as LoRA-MoE [21], could make our approach viable for resource-constrained environments [43].

Furthermore, there is considerable potential in exploring expert-distributed deployment, where $\mathcal{L}_{balance}$ can optimize both parameter inference efficiency and model performance. This avenue could significantly enhance the scalability and practicality of MoE models in real-world applications, providing new opportunities for distributed expert architectures.

## 7 Conclusion

In this work, we present a theoretically grounded framework that resolves the inherent conflict between expert specialization and routing uniformity in MoE training. By introducing orthogonality and variance-based objectives, our method significantly improves downstream performance without any architectural changes. This demonstrates that MoE efficiency and specialization can be simultaneously optimized through loss-level innovations alone. Experiments show the effectiveness of our method.

## 8 Acknowledgements

This work was supported in part by the National Key Research and Development Program of China under Grant 2022YFB2902200; in part by the Guangxi Key Research and Development Program under Grant FN2504240005; in part by the National Natural Science Foundation of China under Grant 62471064; in part by the Fundamental Research Funds for the Beijing University of Posts and Telecommunications under Grant 2025AI4S02.

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

# A  Notations

Table 2: Notations and Definitions

| Notation | Definition |
|----------|------------|
| $\mathcal{L}$ | Total loss function. |
| $\mathcal{L}_h$ | Primary task loss. |
| $\mathcal{L}_{aux}$ | Auxiliary loss function. |
| $\mathcal{L}_o$ | Orthogonality loss. |
| $\mathcal{L}_v$ | Variance loss. |
| $x_i$ | A d-dimensional input token vector, $x_i \in \mathbb{R}^d$. |
| $N$ | Number of tokens in a sequence or batch. |
| $E_j$ | The j-th expert network. |
| $\theta_{E_j}$ | Parameters of the j-th expert network $E_j$. |
| $h(x_i)$ | Vector of logits output by the routing network for token $x_i$, $h(x_i) \in \mathbb{R}^n$. |
| $h(x_i)_j$ | The j-th component of the logit vector $h(x_i)$, corresponding to expert $j$. |
| $P(x_i)_j$ | Initial routing probability of token $x_i$ for expert $E_j$. |
| $T_i$ | Set of indices of the top k experts selected for token $x_i$. |
| $s_{ij}$ | Final routing weight assigned to the i-th token for the j-th expert. |
| $y_i$ | Final output for token $x_i$ from the MoE layer. |
| $E_j(x_i)$ | Output of expert $E_j$ for token $x_i$. |
| $f_j$ | Proportion of tokens assigned to expert j. |
| $p_j$ | Sum of routing probabilities (scores) assigned to expert j across all N tokens in a batch, $p_j = \sum_{i=1}^{N} s_{ij}$. |
| $\mathbb{I}_{\{s_{ij} > \tau_{gate}\}}$ | Indicator function ensuring $\tilde{x}_{ij}$ is $E_j(x_i)$ if $s_{ij} > \tau_{gate}$ and zero otherwise. |
| $\theta_R$ | Parameters of the routing network. |
| $W_{ij}(\theta_R)$ | Raw logit produced by the routing network for token $x_i$ and expert j. |
| $s'_{ij}$ | Soft routing probabilities obtained via a softmax function applied to logits $W_{ij}(\theta_R)$. |
| $E_{avg}(x_i)$ | Approximate average output of experts for token $x_i$ when experts become similar. |
| $\tilde{x}_{ij}$ | Output of expert $E_j$ for token $x_i$ if $s_{ij} > \tau_{gate}$, zero vector otherwise; $\tilde{x}_{ij} = E_j(x_i) \cdot \mathbb{I}_{\{s_{ij} > \tau_{gate}\}}$. |
| $\tau_{gate}$ | Threshold for routing score $s_{ij}$ to consider an expert active for orthogonality loss calculation. |
| $\epsilon_{norm}$ | Small constant added to the denominator in orthogonality loss to prevent division by zero. |
| $proj_{\tilde{x}_{ik}}(\tilde{x}_{ij})$ | Vector projection of $\tilde{x}_{ij}$ onto $\tilde{x}_{ik}$. |

# B  Motivation

## B.1  MoE Layer Structure

A Mixture of Experts (MoE) layer enhances the capacity of a neural network model by conditionally activating different specialized sub-networks, known as "experts," for different input tokens. This architecture allows the model to scale its parameter count significantly while maintaining a relatively constant computational cost per token during inference.

Let the input to the MoE layer be a sequence of $N$ tokens, denoted as $X = \{x_1, x_2, \ldots, x_N\}$, where each token $x_i \in \mathbb{R}^d$ is a $d$-dimensional vector. The MoE layer comprises a set of $n$ independent expert networks, $E = \{E_1, E_2, \ldots, E_n\}$. Each expert $E_j$ is typically a feed-forward network (FFN) with its own set of parameters $\theta_{E_j}$.

A crucial component of the MoE layer is the routing network, also known as the gating network, $G$. The routing network takes an input token $x_i$ and determines which experts should process this token. It outputs a vector of logits $h(x_i) \in \mathbb{R}^n$, where each component $h(x_i)_j$ corresponds to the $j$-th expert. These logits are then typically passed through a softmax function to produce initial routing probabilities or scores:

$$P(x_i)_j = \frac{\exp(h(x_i)_j)}{\sum_{k=1}^n \exp(h(x_i)_k)}, \quad \text{for } j = 1, \ldots, n. \tag{11}$$

These probabilities $P(x_i)_j$ represent the initial affinity of token $x_i$ for expert $E_j$.

To manage computational cost and encourage specialization, a top-k selection mechanism is often employed. For each token $x_i$, the top $k$ experts (where $k \ll n$, often $k = 1$ or $k = 2$) with the highest routing probabilities $P(x_i)_j$ are chosen. Let $T_i \subset \{1, \ldots, n\}$ be the set of indices of the top $k$ experts selected for token $x_i$. The routing scores are then re-normalized or directly used based on this selection. The routing score matrix $\mathcal{S}$ of dimensions $N \times n$ captures these assignments:

$$\mathcal{S} = \begin{pmatrix} s_{11} & s_{12} & \cdots & s_{1n} \\ s_{21} & s_{22} & \cdots & s_{2n} \\ \vdots & \vdots & \ddots & \vdots \\ s_{N1} & s_{N2} & \cdots & s_{Nn} \end{pmatrix}, \tag{12}$$

where $s_{ij}$ represents the final weight assigned to the $i$-th token for the $j$-th expert. If expert $j$ is among the top $k$ selected for token $x_i$ (i.e., $j \in T_i$), then $s_{ij}$ is typically derived from $P(x_i)_j$ (e.g., by re-normalizing the top-k probabilities so they sum to 1, or simply $s_{ij} = P(x_i)_j / \sum_{l \in T_i} P(x_i)_l$). If expert $j$ is not selected for token $x_i$ (i.e., $j \notin T_i$), then $s_{ij} = 0$. Consequently, for each token $x_i$, the sum of its routing scores across all experts is normalized:

$$\sum_{j=1}^n s_{ij} = 1, \quad \text{for } i = 1, 2, \ldots, N. \tag{13}$$

It is important to note that with a top-k mechanism where $k < n$, most $s_{ij}$ values for a given $i$ will be zero.

Each token $x_i$ is then processed by its selected experts. The output of expert $E_j$ for token $x_i$ is denoted as $E_j(x_i)$. The final output $y_i$ for token $x_i$ from the MoE layer is a weighted sum of the outputs from all experts, using the routing scores as weights:

$$y_i = \sum_{j=1}^n s_{ij} E_j(x_i). \tag{14}$$

Since $s_{ij} = 0$ for non-selected experts, this sum is effectively only over the top $k$ chosen experts for token $x_i$.

To encourage a balanced load across the experts and prevent a situation where only a few experts are consistently chosen (expert starvation), an auxiliary loss function, $\mathcal{L}_{aux}$, is commonly introduced. Let $F = \{f_1, f_2, \ldots, f_n\}$ represent the proportion of tokens assigned to each expert. More precisely, $f_j$ can be defined as the fraction of tokens in a batch for which expert $j$ is among the top $k$ selected experts, or it can be a softer measure. For a given MoE layer, the total loss function $\mathcal{L}$ consists of two main parts: the primary task loss $\mathcal{L}_h$ (e.g., cross-entropy loss in language modeling) and the auxiliary loss $\mathcal{L}_{aux}$:

$$\mathcal{L} = \mathcal{L}_h + \alpha \cdot \mathcal{L}_{aux}. \tag{15}$$

Here, $\mathcal{L}_h$ is computed based on the final output $Y = \{y_1, y_2, \ldots, y_N\}$ of the MoE layer (and subsequent layers), and $\alpha$ is a scalar hyperparameter that controls the importance of the auxiliary loss term. The auxiliary loss is often designed to penalize imbalance in the distribution of tokens to experts. A common formulation for $\mathcal{L}_{aux}$, as referenced in the original text, involves the sum of routing scores per expert:

$$\mathcal{L}_{aux} = \sum_{j=1}^n (\text{load}_j \cdot \text{importance}_j), \tag{16}$$

where $\text{load}_j$ is related to the number of tokens routed to expert $j$, and $\text{importance}_j$ is related to the routing probabilities for expert $j$. Let $p_j$ represent the sum of routing probabilities (scores) assigned to expert $j$ across all $N$ tokens in the batch:

$$p_j = \sum_{i=1}^{N} s_{ij}. \tag{17}$$

This $p_j$ value gives an indication of the "total routing score" directed towards expert $j$. The term $f_j$ in the original formulation, representing the proportion of tokens assigned to expert $j$, can be considered as the average routing probability for expert $j$ over the batch, i.e., $f_j = \frac{1}{N} \sum_{i=1}^{N} \mathbb{I}(j \in T_i)$, where $\mathbb{I}(\cdot)$ is the indicator function, or a softer version using $s_{ij}$. The specific form $\mathcal{L}_{aux} = \sum_{j=1}^{n} f_j \cdot p_j$ as given in the prompt, if $f_j$ is interpreted as an average probability or fraction of tokens assigned, and $p_j$ is the sum of probabilities, then $f_j \cdot p_j$ would be $(\frac{1}{N} \sum_i s_{ij}) \cdot (\sum_i s_{ij})$. However, a more standard auxiliary loss aims to balance the load, often by taking the form of the dot product of the vector of the fraction of tokens dispatched to each expert and the vector of the fraction of router probability dispatched to each expert, scaled by the number of experts. For example, a common auxiliary load balancing loss used in literature (e.g., Switch Transformers) is:

$$\mathcal{L}_{aux} = \alpha \cdot n \sum_{j=1}^{n} \left( \frac{1}{N} \sum_{i=1}^{N} \mathbb{I}(j \in T_i) \right) \cdot \left( \frac{1}{N} \sum_{i=1}^{N} P(x_i)_j \right), \tag{18}$$

or using $s_{ij}$ values directly related to $P(x_i)_j$ for the selected experts. The intent is to make the product of the actual load (how many tokens an expert gets) and the routing confidence for that expert more uniform across experts. If $f_j$ in the original text refers to $N_j/N$ (fraction of tokens routed to expert $j$) and $p_j$ is $\sum_{i=1}^{N} s_{ij}$ (sum of gating values for expert $j$ over the batch, which already considers the top-k selection implicitly through $s_{ij}$), then the formula from the prompt:

$$\mathcal{L} = \mathcal{L}_h + \alpha \sum_{j=1}^{n} f_j \cdot p_j \tag{19}$$

where $f_j$ is the proportion of tokens assigned to expert $j$, and $p_j = \sum_{i=1}^{N} s_{ij}$ is the sum of routing weights for expert $j$. This auxiliary loss encourages the gating network to distribute tokens such that experts with higher $p_j$ (receiving larger aggregate routing weights) are also assigned a substantial fraction of tokens $f_j$, aiming for a balance in expert utilization.

## B.2 Observation

***Obs I(Expert Overlap)**: Introduction of the auxiliary loss function leads to a more homogenized distribution of tokens across experts, which may reduce the distinctiveness of each expert.*

It has been observed that the auxiliary loss function is independent of the expert parameter matrices $\theta_{E_j}$. Therefore, for the $j$-th expert, its gradient can be written as:

$$\frac{\partial \mathcal{L}}{\partial \theta_{E_j}} = \frac{\partial \mathcal{L}_h}{\partial \theta_{E_j}} + \alpha \cdot \frac{\partial \mathcal{L}_{aux}}{\partial \theta_{E_j}} = \frac{\partial \mathcal{L}}{\partial y_h} \cdot \frac{\partial y_h}{\partial \theta_{E_j}} = \sum_{i=1}^{N} x_i \cdot s_{ij}, j = 1, 2, \cdots, n. \tag{20}$$

where $\theta_{E_j}$ is the parameter matrix of the $j$-th expert, and $y_h$ is the output of the MoE layer. During gradient descent, the addition of the auxiliary loss $\mathcal{L}_{aux}$ forces the routing mechanism to evenly distribute the tokens across experts as much as possible. This results in input token $x_i$ being assigned to an expert that may not be semantically aligned with it, causing an unintended gradient flow to expert $j$. Mathematically, after applying the top-k mechanism, the routing score $s_{ij}$ transitions from 0 to a non-zero value, introducing gradients from tokens that originally had no affinity with expert $j$.

***Obs II(Routing Uniformity)**: As training progresses, the routing output tends to become more uniform, with the expert weight distribution gradually converging towards an equal allocation.*

To understand this phenomenon, we first examine the source of gradients with respect to the routing parameters $\theta_R$. Let $W_{ij}(\theta_R)$ denote the raw logit produced by the routing network for token $x_i$ and

expert $j$. The soft routing probabilities, denoted as $s'_{ij}$, are typically obtained via a softmax function applied to these logits:

$$s'_{ij} = \frac{\exp(W_{ij}(\theta_R))}{\sum_{k=1}^{n} \exp(W_{ik}(\theta_R))}. \tag{21}$$

These soft probabilities $s'_{ij}$ are then used to determine the final routing assignments $s_{ij}$ in the matrix $\mathcal{S}$ (after top-k selection). The derivatives $\frac{\partial s_{ij}}{\partial \theta_R}$ in the gradient expressions are understood to represent the differentiation through these underlying soft probabilities with respect to the router parameters $\theta_R$. The total loss $\mathcal{L}$ comprises the main task loss $\mathcal{L}_h$ and the auxiliary loss $\mathcal{L}_{aux}$. The gradient of $\mathcal{L}$ with respect to $\theta_R$ is given by:

$$\frac{\partial \mathcal{L}}{\partial \theta_R} = \frac{\partial \mathcal{L}_h}{\partial \theta_R} + \alpha \cdot \frac{\partial \mathcal{L}_{aux}}{\partial \theta_R}. \tag{22}$$

Substituting the expressions provided in the context, we have:

$$\frac{\partial \mathcal{L}}{\partial \theta_R} = \sum_{i=1}^{N} \left( \sum_{j=1}^{n} (x_i \cdot \theta_{E_j}) \frac{\partial s_{ij}}{\partial \theta_R} \right) + \alpha \cdot \sum_{j=1}^{n} f_j \sum_{i=1}^{N} \frac{\partial s_{ij}}{\partial \theta_R}, \tag{23}$$

where $x_i \cdot \theta_{E_j}$ represents the output of expert $j$ for token $x_i$, and $f_j$ denotes the fraction of tokens ultimately assigned to expert $j$.

The first term, $\frac{\partial \mathcal{L}_h}{\partial \theta_R} = \sum_{i=1}^{N} \sum_{j=1}^{n} (x_i \cdot \theta_{E_j}) \frac{\partial s_{ij}}{\partial \theta_R}$, represents the gradient contribution from the main task loss. This term guides the router to select experts that are most beneficial for minimizing $\mathcal{L}_h$. However, as discussed in *Obs I*, the expert parameters $\theta_{E_j}$ tend to become similar during training due to overlapping token assignments induced by $\mathcal{L}_{aux}$. Consequently, the expert outputs $x_i \cdot \theta_{E_j}$ become less distinguishable across different experts $j$ for a given token $x_i$. Let $x_i \cdot \theta_{E_j} \approx E_{\text{avg}}(x_i)$ for all $j$. In this scenario, the specific choice of expert $j$ (i.e., making $s_{ij}$ large for that $j$) has a progressively similar impact on $\mathcal{L}_h$, regardless of which $j$ is chosen. The differential information $(x_i \cdot \theta_{E_j}) - (x_i \cdot \theta_{E_k})$ between experts diminishes. As a result, the router receives a weaker, less discriminative signal from the main loss component for selecting specific experts. The ability of $\mathcal{L}_h$ to guide fine-grained, specialized routing decisions is therefore reduced.

With the diminishing influence of $\frac{\partial \mathcal{L}_h}{\partial \theta_R}$, the updates to the routing parameters $\theta_R$ become increasingly dominated by the auxiliary loss gradient, $\alpha \frac{\partial \mathcal{L}_{aux}}{\partial \theta_R}$:

$$\frac{\partial \mathcal{L}}{\partial \theta_R} \approx \alpha \frac{\partial \mathcal{L}_{aux}}{\partial \theta_R} = \alpha \sum_{j=1}^{n} f_j \sum_{i=1}^{N} \frac{\partial s_{ij}}{\partial \theta_R}. \tag{24}$$

The auxiliary loss $\mathcal{L}_{aux}$ is designed to encourage a balanced load across experts, primarily by promoting uniformity in $f_j$ (the fraction of tokens processed by expert $j$). This is achieved by using $p_j = \sum_{i=1}^{N} s_{ij}$ (where $s_{ij}$ are the post-top-k scores) as a differentiable surrogate to guide the optimization. The objective is to drive $f_j \to 1/n$ for all $n$ experts. The gradient term $\alpha \frac{\partial \mathcal{L}_{aux}}{\partial \theta_R}$ adjusts the router parameters $\theta_R$ (and thus the soft probabilities $s'_{ij}$ which determine $s_{ij}$) to achieve this balanced distribution.

In the absence of strong, discriminative signals from $\mathcal{L}_h$ (due to expert similarity), and under the primary influence of $\mathcal{L}_{aux}$ which penalizes load imbalance, the router tends to adopt a strategy that most straightforwardly achieves load balance. This often results in the soft routing probabilities $s'_{ij}$ for a given token $x_i$ becoming more uniform across the experts $j$, i.e., $s'_{ij} \to 1/n$. If the router assigns nearly equal soft probabilities to all experts for any given token, then the post-top-k scores $s_{ij}$ will also reflect this reduced selectivity, and the sum $p_j = \sum_i s_{ij}$ will naturally tend towards $N/n$, satisfying the auxiliary loss's objective. This trend leads to the variance of routing weights for a given token $x_i$ (i.e., $\text{Var}_j(s'_{ij})$) decreasing over time. Consequently, the overall routing output becomes more uniform, and the router becomes less specialized in its assignments, reinforcing the homogenization observed. This feedback loop, where expert similarity weakens task-specific routing signals and strengthens the homogenizing effect of the load balancing mechanism, explains the progressive trend towards routing uniformity.

## C Method

### C.1 Specialized Losses $\mathcal{L}_o$ and $\mathcal{L}_v$

In this section, we introduce two critical loss functions: the orthogonality loss $\mathcal{L}_o$, which acts on the expert representations, and the variance loss $\mathcal{L}_v$, which acts on the routing scores. These losses are designed to encourage expert specialization and routing diversity, respectively.

**Expert Specialization via Orthogonality Loss $\mathcal{L}_o$.** To foster expert specialization, we aim to make the representations learned by different experts for the same input token as independent as possible. Orthogonal vectors are the epitome of linear independence. Thus, we introduce an orthogonality objective that penalizes similarities between the output representations of different experts when they are selected to process the same token. This is achieved through the orthogonality loss $\mathcal{L}_o$.

The orthogonality loss is defined as:

$$\mathcal{L}_o = \sum_{i=1}^{N} \sum_{j=1}^{n} \sum_{\substack{k=1 \\ k \neq j}}^{n} \frac{\langle \tilde{x}_{ij}, \tilde{x}_{ik} \rangle}{\langle \tilde{x}_{ik}, \tilde{x}_{ik} \rangle + \epsilon_{norm}} \tilde{x}_{ik}, \quad \text{where} \quad \tilde{x}_{ij} = E_j(x_i) \cdot \mathbb{I}_{\{s_{ij} > \tau_{gate}\}}. \quad (25)$$

Here, $N$ is the number of tokens in a batch, and $n$ is the total number of experts. The input token is denoted by $x_i$. The term $E_j(x_i)$ represents the output vector of the $j$-th expert, $E_j$, when processing token $x_i$. The indicator function $\mathbb{I}_{\{s_{ij} > \tau_{gate}\}}$ ensures that $\tilde{x}_{ij}$ is the actual output $E_j(x_i)$ if the routing score $s_{ij}$ for expert $j$ and token $x_i$ exceeds a certain threshold $\tau_{gate}$ (implying expert $j$ is selected in the top-$k$ routing for token $x_i$), and $\tilde{x}_{ij}$ is a zero vector otherwise. This effectively means that the loss operates only on the experts that are active for a given token. A small constant $\epsilon_{norm}$ is added to the denominator to prevent division by zero if an expert's output vector happens to be zero.

The core component of $\mathcal{L}_o$, $\text{proj}_{\tilde{x}_{ik}}(\tilde{x}_{ij}) = \frac{\langle \tilde{x}_{ij}, \tilde{x}_{ik} \rangle}{\langle \tilde{x}_{ik}, \tilde{x}_{ik} \rangle + \epsilon_{norm}} \tilde{x}_{ik}$, calculates the vector projection of the output $\tilde{x}_{ij}$ (from expert $j$ for token $i$) onto the output $\tilde{x}_{ik}$ (from expert $k$ for the same token $i$). The loss $\mathcal{L}_o$ sums these projection vectors for all distinct pairs of active experts $(j, k)$ for each token $x_i$, and then sums these across all tokens in the batch.

Although the formula (25) presents $\mathcal{L}_o$ as a sum of vectors, the optimization objective is to minimize the magnitude of these projection components. Typically, this is achieved by minimizing a scalar value derived from these vectors, such as the sum of their squared $L_2$ norms, i.e., $\sum_{i,j,k \neq j} \|\text{proj}_{\tilde{x}_{ik}}(\tilde{x}_{ij})\|^2$. Minimizing these projections encourages the dot product $\langle \tilde{x}_{ij}, \tilde{x}_{ik} \rangle$ to approach zero for $j \neq k$. This forces the representations $\tilde{x}_{ij}$ and $\tilde{x}_{ik}$ from different active experts to become more orthogonal.

By minimizing $\mathcal{L}_o$, we reduce the representational overlap between different experts chosen for the same token. This encourages each expert to learn unique features or specialize in processing different aspects of the input data, leading to a more diverse and efficient set of experts. This specialization is key to mitigating the expert overlap issue.

**Routing Diversification via Variance Loss $\mathcal{L}_v$.** To ensure that the router utilizes experts in a varied and balanced manner, rather than consistently favoring a few, we introduce a variance-based loss $\mathcal{L}_v$. This loss encourages the routing scores assigned by the router to be more diverse across tokens for any given expert.

The variance loss is defined as:

$$\mathcal{L}_v = -\sum_{i=1}^{N} \sum_{j=1}^{n} \frac{1}{n} \cdot (s_{ij} - \bar{s}_j)^2, \quad \text{where} \quad \bar{s}_j = \frac{1}{N} \cdot \sum_{i=1}^{N} s_{ij}. \quad (26)$$

In this formula, $s_{ij}$ represents the routing score (e.g., gating value from a softmax layer in the router) indicating the router's preference for assigning token $x_i$ to expert $E_j$. The term $\bar{s}_j$ is the average routing score for expert $E_j$ calculated across all $N$ tokens in the current batch. This average score, $\bar{s}_j$, can be interpreted as a measure of the current utilization or overall assignment strength for expert $j$ within that batch.

The core of the loss, $(s_{ij} - \bar{s}_j)^2$, measures the squared deviation of the specific score $s_{ij}$ from the average score $\bar{s}_j$ for expert $j$. A sum of these squared deviations for a particular expert $j$ over all tokens, $\sum_{i=1}^{N} (s_{ij} - \bar{s}_j)^2$, quantifies the total variance of routing scores received by that expert. A

high variance implies that expert $j$ receives a wide range of scores from different tokens (i.e., it is strongly preferred for some tokens and weakly for others), rather than receiving similar scores for all tokens it processes.

The loss $\mathcal{L}_v$ sums these squared deviations over all experts $j$ (scaled by $1/n$) and all tokens $i$, and then negates this sum. Therefore, minimizing $\mathcal{L}_v$ is equivalent to maximizing the sum of these score variances: $\sum_{j=1}^{n} \sum_{i=1}^{N} (s_{ij} - \bar{s}_j)^2$. This maximization encourages the routing mechanism to produce a diverse set of scores for each expert across different tokens.

By promoting higher variance in routing scores per expert, $\mathcal{L}_v$ helps to prevent routing uniformity, where experts might be selected with similar probabilities for many tokens or where some experts are consistently overloaded while others are underutilized based on uniform high/low scores. Instead, it pushes the router to make more discriminative assignments, which can lead to better load balancing in conjunction with $\mathcal{L}_{aux}$ and supports experts in specializing on more distinct subsets of tokens.

## C.2 Compatibility of Multi-Objective Optimization

In this section, we conduct a detailed analysis of how each loss component, namely $\mathcal{L}_h, \mathcal{L}_{aux}, \mathcal{L}_o, \mathcal{L}_v$, influences the optimization dynamics of expert parameters $\theta_{E_j}$ (for $j = 1, \ldots, n$ experts) and routing parameters $\theta_R$ during the training process. Our primary focus is to demonstrate the theoretical compatibility and synergistic interplay between the specialized losses $\mathcal{L}_o$ (promoting expert orthogonality) and $\mathcal{L}_v$ (promoting routing score diversification) in conjunction with the load balancing loss $\mathcal{L}_{aux}$ and the primary task loss $\mathcal{L}_h$. The analysis is structured around two key questions:

***Balancing Expert and Routing.*** *How can expert ($\mathcal{L}_o$) and routing ($\mathcal{L}_v$) optimizations be designed to complement each other without compromising their respective objectives, and how do they interact with $\mathcal{L}_{aux}$?*

We begin by demonstrating that the optimization objectives $\mathcal{L}_o$ and $\mathcal{L}_v$ are compatible in their optimization directions with respect to the expert parameters $\theta_{E_j}$ and routing parameters $\theta_R$. Subsequently, we will show that these losses can mutually reinforce each other, leading to a more effective and stable learning process for Mixture-of-Experts (MoE) models.

**Mutually Compatible**

We elaborate on the compatibility of $\mathcal{L}_o$ and $\mathcal{L}_v$ by examining their respective gradient contributions to expert parameters and routing parameters. The total loss function is $\mathcal{L} = \mathcal{L}_h + \alpha \mathcal{L}_{aux} + \beta \mathcal{L}_o + \gamma \mathcal{L}_v$.

From the **expert parameter $\theta_{E_j}$ perspective**, the expert parameters $\theta_{E_j}$ are primarily updated to minimize the task loss $\mathcal{L}_h$ for the tokens routed to expert $j$, and to satisfy the orthogonality constraint $\mathcal{L}_o$. The auxiliary loss $\mathcal{L}_{aux}$ and the variance loss $\mathcal{L}_v$ are functions of the routing scores $s_{ij}$ (outputs of the router $R(x_i)_{\theta_R}$) and do not explicitly depend on the expert parameters $\theta_{E_j}$. That is, $\frac{\partial \mathcal{L}_{aux}}{\partial \theta_{E_j}} = 0$ and $\frac{\partial \mathcal{L}_v}{\partial \theta_{E_j}} = 0$. The output of expert $j$ for token $x_i$ is denoted as $\tilde{x}_{ij} = E_j(x_i; \theta_{E_j}) \cdot \mathbb{I}_{\{s_{ij}>0\}}$, where $E_j(x_i; \theta_{E_j})$ is the transformation by expert $j$ (e.g., $x_i \theta_{E_j}$ if $x_i$ is a row vector and $\theta_{E_j}$ is a weight matrix), and $\mathbb{I}_{\{s_{ij}>0\}}$ is an indicator function that is 1 if $x_i$ is routed to expert $j$ (i.e., $s_{ij}$ is among the top-$k$ scores for $x_i$) and 0 otherwise. For simplicity in gradient derivation with respect to $\theta_{E_j}$, we consider only tokens $x_i$ for which expert $j$ is active. The gradient of the total loss $\mathcal{L}$ with respect to $\theta_{E_j}$ is:

$$\frac{\partial \mathcal{L}}{\partial \theta_{E_j}} = \sum_{i=1}^{N} \mathbb{I}_{\{s_{ij}>0\}} \left( \frac{\partial \mathcal{L}_h}{\partial \tilde{x}_{ij}} + \beta \frac{\partial \mathcal{L}_o}{\partial \tilde{x}_{ij}} \right) \frac{\partial \tilde{x}_{ij}}{\partial \theta_{E_j}}. \tag{27}$$

Let $g_{y_i} = \frac{\partial \mathcal{L}_h}{\partial y_i}$ be the gradient of the task loss with respect to the final output $y_i = \sum_k s_{ik} \tilde{x}_{ik}$. Then $\frac{\partial \mathcal{L}_h}{\partial \tilde{x}_{ij}} = g_{y_i} s_{ij}$. The orthogonality loss $\mathcal{L}_o$ is designed to make $\tilde{x}_{ij}$ and $\tilde{x}_{ik}$ (for $k \neq j$, $k$ also selected for $x_i$) orthogonal. Assuming the specific form of $\mathcal{L}_o$ from the paper leads to the gradient component shown (interpreted as $\frac{\partial \mathcal{L}_o}{\partial \tilde{x}_{ij}}$ contributing $\sum_{k=1, k \neq j}^{n} \frac{\tilde{x}_{ik} \tilde{x}_{ik}^\top}{\langle \tilde{x}_{ik}, \tilde{x}_{ik} \rangle} \tilde{x}_{ij}$), and if $\frac{\partial \tilde{x}_{ij}}{\partial \theta_{E_j}}$ results in a factor of $x_i^T$

(assuming $\tilde{x}_{ij} = \theta_{E_j} x_i$ with $x_i$ as column vector), the gradient expression given in the paper is:

$$\frac{\partial \mathcal{L}}{\partial \theta_{E_j}} = \sum_{i=1}^{N} \mathbb{I}_{\{s_{ij}>0\}} \left( g_{y_i} s_{ij} + \beta \left( \sum_{\substack{k=1 \\ k \neq j \\ \mathbb{I}_{\{s_{ik}>0\}}}}^{n} \frac{\tilde{x}_{ik}\tilde{x}_{ik}^{\top}}{\langle \tilde{x}_{ik}, \tilde{x}_{ik} \rangle} \tilde{x}_{ij} \right) \right) x_i^T. \tag{28}$$

More generally, using the paper's notation for the gradient w.r.t. $\theta_{E_j}$ directly:

$$\frac{\partial \mathcal{L}}{\partial \theta_{E_j}} = \sum_{i=1}^{N} \left( \underbrace{g_{\mathcal{L}_h}(\tilde{x}_{ij}, s_{ij})}_{\text{from } \mathcal{L}_h} + \beta \cdot \underbrace{g_{\mathcal{L}_o}(\{\tilde{x}_{il}\}_{l \neq j}, \tilde{x}_{ij})}_{\text{from } \mathcal{L}_o} \right) \frac{\partial \tilde{x}_{ij}}{\partial \theta_{E_j}}, \tag{29}$$

where $g_{\mathcal{L}_h}(\tilde{x}_{ij}, s_{ij})$ represents the gradient contribution from $\mathcal{L}_h$ to $\tilde{x}_{ij}$ (e.g., $s_{ij}$ in the paper's simplified notation might represent $g_{y_i} s_{ij}$ or a similar term) and $g_{\mathcal{L}_o}(\{\tilde{x}_{il}\}_{l \neq j}, \tilde{x}_{ij})$ represents the gradient contribution from $\mathcal{L}_o$ (e.g., $\sum_{\substack{k=1 \\ k \neq j}}^{n} \frac{\tilde{x}_{ik}\tilde{x}_{ik}^{\top}}{\langle \tilde{x}_{ik}, \tilde{x}_{ik} \rangle} \tilde{x}_{ij}$ if it acts on $\tilde{x}_{ij}$). The crucial observation is that $\mathcal{L}_{aux}$ and $\mathcal{L}_v$ do not directly impose conflicting gradient directions on $\theta_{E_j}$ as their influence is on $\theta_R$. As training progresses, $\mathcal{L}_o$ encourages $\theta_{E_j}$ to form specialized representations. This specialization, driven by $\mathcal{L}_o$, is not hindered by $\mathcal{L}_{aux}$ or $\mathcal{L}_v$.

From the **routing parameter $\theta_R$ perspective**, the routing parameters $\theta_R$ determine the routing scores $s_{ij} = R(x_i, j; \theta_R)$. The gradient of the total loss with respect to $\theta_R$ is given by:

$$\frac{\partial \mathcal{L}}{\partial \theta_R} = \sum_{i=1}^{N} \sum_{j=1}^{n} \frac{\partial \mathcal{L}}{\partial s_{ij}} \frac{\partial s_{ij}}{\partial \theta_R}. \tag{30}$$

The term $\frac{\partial \mathcal{L}}{\partial s_{ij}}$ captures influences from all relevant loss components:

$$\frac{\partial \mathcal{L}}{\partial s_{ij}} = \frac{\partial \mathcal{L}_h}{\partial s_{ij}} + \alpha \frac{\partial \mathcal{L}_{aux}}{\partial s_{ij}} + \beta \frac{\partial \mathcal{L}_o}{\partial s_{ij}} + \gamma \frac{\partial \mathcal{L}_v}{\partial s_{ij}}. \tag{31}$$

The paper asserts that $\mathcal{L}_o$ does not directly affect the gradient with respect to routing parameters $\theta_R$, implying $\frac{\partial \mathcal{L}_o}{\partial s_{ij}} = 0$. This holds if $\mathcal{L}_o$ is defined based on the expert outputs $\tilde{x}_{ij}$ which, once an expert is selected, depend on $\theta_{E_j}$ and $x_i$ but not on the magnitude of $s_{ij}$ itself (assuming $s_{ij}$ is used for hard selection via top-k, and not as a differentiable weighting for $\tilde{x}_{ij}$ within $\mathcal{L}_o$'s definition). Given this assumption, the gradient $\frac{\partial \mathcal{L}}{\partial s_{ij}}$ becomes:

$$\frac{\partial \mathcal{L}}{\partial s_{ij}} = \underbrace{g_{y_i}^T \tilde{x}_{ij}}_{\text{from } \mathcal{L}_h} + \alpha \underbrace{\frac{\partial \mathcal{L}_{\text{aux}}}{\partial s_{ij}}}_{\text{from } \mathcal{L}_{\text{aux}}} + \gamma \underbrace{\frac{\partial \mathcal{L}_v}{\partial s_{ij}}}_{\text{from } \mathcal{L}_v}. \tag{32}$$

Substituting the specific forms for derivatives of $\mathcal{L}_{aux}$ and $\mathcal{L}_v$ (where $\mathcal{L}_{aux}$ often involves balancing the load $f_j = \sum_i s_{ij}/N$ or similar, and $\mathcal{L}_v = -\sum_{i=1}^{N} \sum_{j=1}^{n} \frac{1}{n} \cdot (s_{ij} - \bar{s}_j)^2$), the paper's specific form for the gradient of the total loss w.r.t. $\theta_R$ is:

$$\frac{\partial \mathcal{L}}{\partial \theta_R} = \sum_{i=1}^{N} \sum_{j=1}^{n} \left( \underbrace{g_{y_i}^T \tilde{x}_{ij}}_{\text{term from } \mathcal{L}_h} + \alpha \cdot \underbrace{\text{derived from } f_j}_{\text{term from } \mathcal{L}_{aux}} - \gamma \cdot \underbrace{\frac{2(N-1)}{nN} \cdot (s_{ij} - \bar{s}_j)}_{\text{term from } \mathcal{L}_v} \right) \cdot \frac{\partial s_{ij}}{\partial \theta_R}. \tag{33}$$

The term represented by $\tilde{x}_{ij}$ in the paper's original routing gradient formula corresponds to $g_{y_i}^T \tilde{x}_{ij}$, $f_j$ to the derivative of $\mathcal{L}_{aux}$, and the last term to the derivative of $\mathcal{L}_v$. This gradient is influenced by the expert representations $\tilde{x}_{ij}$ (via $\mathcal{L}_h$), the expert load $f_j$ (via $\mathcal{L}_{aux}$), and the distribution of routing weights $s_{ij}$ (via $\mathcal{L}_v$). The optimization of $\mathcal{L}_v$ aims to diversify routing scores, while $\mathcal{L}_{aux}$ aims to balance loads. These objectives are not inherently contradictory with the primary task of minimizing $\mathcal{L}_h$. For instance, $\mathcal{L}_v$ might encourage a token to be strongly assigned to one expert within its top-k set, while $\mathcal{L}_{aux}$ ensures that, across all tokens, experts are utilized in a balanced manner. The absence of a direct gradient from $\mathcal{L}_o$ on $s_{ij}$ (and thus $\theta_R$) prevents direct conflicts between expert orthogonalization and the routing objectives.

Based on this detailed analysis of gradient components, we can summarize:

**Summary.** Expert parameters $\theta_{E_j}$ are updated based on gradients from $\mathcal{L}_h$ and $\mathcal{L}_o$. The losses $\mathcal{L}_{aux}$ and $\mathcal{L}_v$ do not directly contribute gradients to $\theta_{E_j}$, thus avoiding conflicts with expert specialization. Routing parameters $\theta_R$ are influenced by gradients from $\mathcal{L}_h$, $\mathcal{L}_{aux}$, and $\mathcal{L}_v$. The objective of $\mathcal{L}_o$ (expert orthogonality) does not directly impose constraints on $\theta_R$, and the objectives of $\mathcal{L}_{aux}$ (load balancing) and $\mathcal{L}_v$ (score diversification) are designed to be compatible aspects of routing.

**Mutually Reinforcing**

Beyond mere compatibility, $\mathcal{L}_o$ and $\mathcal{L}_v$ can create a synergistic effect, where improvements in one facilitate the optimization of the other.

The orthogonality loss $\mathcal{L}_o$ encourages the effective output vectors of different selected experts, $\tilde{x}_{ij}$ and $\tilde{x}_{ik}$ (for $j \neq k$ and both $j, k$ selected for token $x_i$), to become more orthogonal, i.e., $\langle \tilde{x}_{ij}, \tilde{x}_{ik} \rangle \approx 0$. The learning signal for the routing mechanism, particularly the part derived from the primary task loss $\mathcal{L}_h$ with respect to the routing score $s_{ij}$, is crucial. This component is given by:

$$\frac{\partial \mathcal{L}_h}{\partial s_{ij}} = \frac{\partial \mathcal{L}_h}{\partial y_i} \frac{\partial y_i}{\partial s_{ij}} = g_{y_i}^T \tilde{x}_{ij}, \quad \text{where } y_i = \sum_k s_{ik} \tilde{x}_{ik} \text{ and } g_{y_i} = \frac{\partial \mathcal{L}_h}{\partial y_i}. \tag{34}$$

The full gradient for $s_{ij}$ (excluding $\mathcal{L}_o$'s direct term as discussed) is:

$$\frac{\partial \mathcal{L}}{\partial s_{ij}} = \underbrace{g_{y_i}^T \tilde{x}_{ij}}_{\text{from } \mathcal{L}_h} + \underbrace{\alpha \frac{\partial \mathcal{L}_{\text{aux}}}{\partial s_{ij}}}_{\text{from } \mathcal{L}_{\text{aux}}} + \underbrace{\gamma \frac{\partial \mathcal{L}_v}{\partial s_{ij}}}_{\text{from } \mathcal{L}_v}. \tag{35}$$

Let $p_{ij} = g_{y_i}^T \tilde{x}_{ij}$ represent the projection of the task gradient $g_{y_i}$ onto the expert output $\tilde{x}_{ij}$. When the expert outputs $\{\tilde{x}_{ij}\}_j$ for a given token $x_i$ tend to be orthogonal, they represent distinct, non-redundant features. For any given task-specific gradient vector $g_{y_i}$, its projections $p_{ij}$ onto these more orthogonal expert output vectors are likely to exhibit greater variance. For example, if $\tilde{x}_{i,j_1}$ and $\tilde{x}_{i,j_2}$ are orthogonal, $g_{y_i}$ might align well with $\tilde{x}_{i,j_1}$ (large $p_{i,j_1}$) but poorly with $\tilde{x}_{i,j_2}$ (small $p_{i,j_2}$). In contrast, if $\tilde{x}_{i,j_1}$ and $\tilde{x}_{i,j_2}$ were nearly collinear, $p_{i,j_1}$ and $p_{i,j_2}$ would likely be very similar. This increased variance in the task-relevant signals $p_{ij}$ provides the routing mechanism with more discriminative information, making it easier to differentiate between the utility of experts for a given token. This, in turn, creates more favorable conditions for $\mathcal{L}_v$, which aims to maximize the variance of routing scores $s_{ij}$, thereby encouraging more decisive routing decisions.

Conversely, $\mathcal{L}_v$ contributes to expert specialization. By promoting diverse routing scores $s_{ij}$, $\mathcal{L}_v$ encourages the router to send different types of tokens to different experts (or to assign tokens with higher confidence to a smaller subset of the top-$k$ experts). This results in each expert $E_j$ being trained on a more specialized subset of tokens, denoted $T_j = \{x_i \mid \text{expert } j \text{ is selected for } x_i \text{ with high score}\}$. As experts see more distinct data distributions, their parameters $\theta_{E_j}$ are more likely to diverge and learn unique features representative of their assigned token subsets $T_j$. This functional divergence naturally promotes the orthogonality of their output representations $\tilde{x}_{ij}$, which is the direct objective of $\mathcal{L}_o$. Thus, $\mathcal{L}_v$ indirectly aids $\mathcal{L}_o$. The statement in the original text "due to the influence of $\mathcal{L}_o$'s gradient $\beta \frac{\partial \mathcal{L}_o}{\partial s_{ij}}$ on $\theta_R$" is interpreted here as an indirect influence: $\mathcal{L}_o$ improves expert representations $\tilde{x}_{ij}$, which in turn makes the routing signal $g_{y_i}^T \tilde{x}_{ij}$ more discriminative, thereby influencing $\theta_R$.

**Summary.** A virtuous cycle is formed: $\mathcal{L}_o$ promotes orthogonal expert outputs $\tilde{x}_{ij}$, which enhances the discriminative power of the routing signals $g_{y_i}^T \tilde{x}_{ij}$. More discriminative routing signals allow $\mathcal{L}_v$ to more effectively diversify routing scores $s_{ij}$. In turn, diversified routing scores $s_{ij}$ driven by $\mathcal{L}_v$ lead to experts being trained on more specialized token subsets, which facilitates the learning of divergent and orthogonal expert parameters $\theta_{E_j}$, thus supporting the objective of $\mathcal{L}_o$. This mutual reinforcement contributes to overall model stability and performance.

***Multi-Objective Optimization.*** *How do expert and routing maintain their balance while enhancing* $\mathcal{L}_{aux}$ *and* $\mathcal{L}_h$ *independently, ensuring mutually beneficial performance improvements?*

The overall objective function $\mathcal{L}$ aims to optimize four key aspects:

1. Accurate data fitting and task performance (minimizing $\mathcal{L}_h$).

2. Orthogonal and specialized expert representations (minimizing $\mathcal{L}_o$).

3. Balanced load distribution across experts (minimizing $\mathcal{L}_{aux}$).

4. Diverse and confident routing decisions (maximizing variance via $\mathcal{L}_v$, i.e., minimizing negative variance).

Our core objective is to achieve an **optimal balance by jointly optimizing these multiple objectives**, ensuring they complement each other for enhanced model performance. The compatibility of these objectives is supported by the following considerations, including the provided lemmata.

**Lemma 1** *Let $\mathcal{S} \in \mathbb{R}^{N \times n}$ be the matrix of routing scores, where $s_{ij}$ is the score for token $i$ assigned to expert $j$. Assume for each token $x_i$ (row of $\mathcal{S}$), $\sum_{j=1}^{n} s_{ij} = 1$ (if scores are normalized probabilities post-softmax) or that $k$ experts are chosen (e.g., $s_{ij} \in \{0, 1/k\}$ or general $s_{ij}$ for selected experts). Then, there always exists a state where the following two objectives are simultaneously optimized: 1. Load balancing: The sum of scores for each expert (column sum, $f_j = \sum_{i=1}^{N} s_{ij}$) tends towards an average value, e.g., $N \cdot k/n$ if each token selects $k$ experts, or $N/n$ if $s_{ij}$ are probabilities and $k = 1$. This is driven by $\mathcal{L}_{aux}$. 2. Routing score variance: For each token $x_i$, the variance of its non-zero routing scores $s_{ij}$ (among the chosen top-$k$ experts) is increased. This is driven by $\mathcal{L}_v$.*

Lemma 1 suggests that the goals of $\mathcal{L}_{aux}$ and $\mathcal{L}_v$ are not inherently contradictory. $\mathcal{L}_{aux}$ focuses on inter-expert load distribution (column-wise property of $\mathcal{S}$), while $\mathcal{L}_v$ focuses on the concentration of routing scores for each token (row-wise property of $\mathcal{S}$). For example, even if each token $x_i$ strongly prefers one expert over others in its top-$k$ set (high variance for $s_{i,:}$), the assignment of tokens to experts can still be managed such that overall expert utilization is balanced. Different tokens can strongly prefer different experts, allowing column sums to balance out.

**Lemma 2** *For two objectives, such as (A) making expert representations orthogonal and (B) balancing the computational load across these experts, it is possible to achieve both. If we consider experts needing to learn distinct "regions" of the problem space (orthogonality) and also needing to process a fair share of the data (load balancing). The original phrasing was: "For two sets of points $\mathcal{A}$ and $\mathcal{B}$ of equal size, it is always possible to partition $\mathcal{A} \cup \mathcal{B}$ such that $\mathcal{A} \cap \mathcal{B} = \varnothing$ and $|\mathcal{A}| = |\mathcal{B}|$." Interpreted in our context: Let $\mathcal{F}_j$ be the functional space or feature set that expert $j$ specializes in. $\mathcal{L}_o$ aims to make $\mathcal{F}_j \cap \mathcal{F}_k = \varnothing$ for $j \neq k$ (orthogonality/specialization). Let $C_j$ be the computational load on expert $j$. $\mathcal{L}_{aux}$ aims to make $C_j \approx C_k$. It is possible for experts to learn distinct specializations ($\mathcal{F}_j$ are disjoint) while still processing a comparable amount of data or tokens ($C_j$ are balanced), provided the data itself contains enough variety to be beneficially partitioned among specialized experts.*

Lemma 2, under this interpretation, suggests that the objectives of expert orthogonalization ($\mathcal{L}_o$) and load balancing ($\mathcal{L}_{aux}$) are compatible. Expert specialization does not necessitate load imbalance, nor does load balance prevent experts from specializing. Each expert can become highly specialized in processing certain types of inputs or learning specific features, while the routing mechanism ensures that the number of tokens processed by each expert remains roughly equal.

In summary, the multi-objective optimization framework is designed for compatibility:

- $\mathcal{L}_{aux}$ and $\mathcal{L}_v$ can be jointly optimized as per Lemma 1.

- $\mathcal{L}_o$ (leading to expert specialization) and $\mathcal{L}_{aux}$ (load balancing) are compatible as per Lemma 2's interpretation.

- As discussed in the "Mutually Reinforcing" section, $\mathcal{L}_o$ and $\mathcal{L}_v$ can synergistically enhance each other. $\mathcal{L}_o$ makes expert outputs more distinct, which helps $\mathcal{L}_v$ by providing clearer signals for routing diversification. Diversified routing, in turn, provides more specialized data streams to experts, aiding $\mathcal{L}_o$.

- All these objectives serve to improve the primary task performance $\mathcal{L}_h$. Specialized experts ($\mathcal{L}_o$) can model complex functions more effectively. Balanced load ($\mathcal{L}_{aux}$) ensures efficient use of resources and prevents undertraining of some experts. Diverse and confident routing ($\mathcal{L}_v$) ensures that tokens are sent to the most appropriate experts.

This comprehensive approach allows the model to harness the benefits of MoE architectures by promoting expert specialization, efficient resource utilization, and decisive routing, all contributing to better overall performance on the downstream task.

## C.3 Proof of Lemmas

> **Lemma 1** *Let $S \in \mathcal{R}^{N \times n}$ be a matrix that satisfies following conditions: each row sums to 1, each row contains $k$ non-zero elements and $n - k$ zero elements. Then, there always exists a state in which the following two objectives are simultaneously optimized: 1. The sum of the elements in each column tends to the average value $\frac{N}{n}$; 2. The variance of the non-zero elements in each row increases.*

### proof C.1  *1. Preliminaries and Assumptions*

*The lemma implicitly requires $k \geq 2$. If $k = 1$, each row $i$ has a single non-zero element $s_{i,j_i} = 1$. The set of non-zero elements for row $i$ is $\{1\}$. Its mean is 1, and its variance is $\frac{1}{1}(1-1)^2 = 0$. This variance cannot be increased as $s_{i,j_i}$ must remain 1. Henceforth, we assume $k \geq 2$.*

*Let $\mathcal{P} = (p_{ij}) \in \{0, 1\}^{N \times n}$ denote the support matrix where $p_{ij} = 1$ if $s_{ij} \neq 0$ and $p_{ij} = 0$ otherwise. Condition (ii) implies $\sum_{j=1}^{n} p_{ij} = k$ for all $i$.*

### 2. Construction of an Initial State $\mathcal{S}^{(0)}$ Optimizing Objective 1

*To optimize Objective 1, we select a support matrix $\mathcal{P}$ such that its column sums (degrees of column nodes in the associated bipartite graph), $d_j = \sum_{i=1}^{N} p_{ij}$, are as uniform as possible. That is, each $d_j \in \{\lfloor Nk/n \rfloor, \lceil Nk/n \rceil\}$. The existence of such a matrix $\mathcal{P}$ is a known result in combinatorics (e.g., provable via network flow arguments or related to the existence of $(0, 1)$-matrices with given marginal sums).*

*Define an initial matrix $\mathcal{S}^{(0)} = (s_{ij}^{(0)})$ based on this $\mathcal{P}$:*

$$s_{ij}^{(0)} = \begin{cases} 1/k & \text{if } p_{ij} = 1 \\ 0 & \text{if } p_{ij} = 0 \end{cases}$$

*This matrix $\mathcal{S}^{(0)}$ satisfies:*

- *Row sums: $\sum_{j=1}^{n} s_{ij}^{(0)} = \sum_{j:p_{ij}=1}(1/k) = k \cdot (1/k) = 1$ for all $i$.*

- *Column sums: $C_j^{(0)} = \sum_{i=1}^{N} s_{ij}^{(0)} = \sum_{i:p_{ij}=1}(1/k) = d_j/k$. Since the integers $d_j$ are as uniform as possible, the values $C_j^{(0)}$ minimize $\sum_{j=1}^{n}(C_j - N/n)^2$. Thus, $\mathcal{S}^{(0)}$ optimizes Objective 1.*

- *Row variance: For any row $i$, the $k$ non-zero elements are all $1/k$. The mean of these non-zero elements is $\mu_i^{(0)} = (1/k)\sum_{j:p_{ij}=1}(1/k) = 1/k$. The variance of these non-zero elements is $Var_i(\mathcal{S}^{(0)}) = \frac{1}{k}\sum_{j:p_{ij}=1}(s_{ij}^{(0)} - \mu_i^{(0)})^2 = \frac{1}{k}\sum_{j:p_{ij}=1}(1/k - 1/k)^2 = 0$.*

### 3. Perturbation via a Cycle in the Support Graph $G_\mathcal{P}$

*Let $G_\mathcal{P} = (U \cup V, E_\mathcal{P})$ be the bipartite graph associated with $\mathcal{P}$, where $U = \{r_1, \ldots, r_N\}$ represents rows, $V = \{c_1, \ldots, c_n\}$ represents columns, and an edge $(r_i, c_j) \in E_\mathcal{P}$ if and only if $p_{ij} = 1$.*

*We assume that for $k \geq 2$, the graph $G_\mathcal{P}$ (corresponding to a $\mathcal{P}$ that optimizes Objective 1 as described above) contains at least one cycle. If $G_\mathcal{P}$ were a forest, this specific perturbation method would not apply. The strength of the lemma's claim ("always exists") suggests that such a cycle is indeed available in an appropriately chosen $\mathcal{P}$.*

*Let such a cycle be $P = (r_1 - c_1 - r_2 - c_2 - \cdots - r_L - c_L - r_1)$. The edges forming this cycle correspond to matrix entries $s_{r_1,c_1}^{(0)}, s_{r_2,c_1}^{(0)}, s_{r_2,c_2}^{(0)}, \ldots, s_{r_1,c_L}^{(0)}$ (indices are re-labeled for cycle elements for simplicity), all of which are equal to $1/k$.*

Define a perturbed matrix $\mathcal{S}' = (s'_{ij})$ by altering elements along this cycle. Let $\delta$ be a scalar such that $0 < \delta \leq 1/k$.

$$s'_{r_1,c_1} = s^{(0)}_{r_1,c_1} + \delta = 1/k + \delta$$
$$s'_{r_2,c_1} = s^{(0)}_{r_2,c_1} - \delta = 1/k - \delta$$
$$s'_{r_2,c_2} = s^{(0)}_{r_2,c_2} + \delta = 1/k + \delta$$
$$\vdots$$
$$s'_{r_L,c_L} = s^{(0)}_{r_L,c_L} + \delta = 1/k + \delta$$
$$s'_{r_1,c_L} = s^{(0)}_{r_1,c_L} - \delta = 1/k - \delta$$

Elements $s'_{ij}$ not involved in the cycle remain $s^{(0)}_{ij}$. Since $\delta \leq 1/k$, all $s'_{ij} \geq 0$. The number of non-zero elements per row remains $k$.

- **Row Sums of $\mathcal{S}'$**: For any row $r_x$ in the cycle (e.g., $r_1$), two of its elements are modified: $s'_{r_1,c_1}$ by $+\delta$ and $s'_{r_1,c_L}$ by $-\delta$. All other non-zero elements in row $r_1$ are unchanged. Thus, the sum $\sum_j s'_{r_1,j} = \sum_j s^{(0)}_{r_1,j} = 1$. This holds for all rows $r_1, \ldots, r_L$. Rows not in the cycle are unaffected.

- **Column Sums of $\mathcal{S}'$**: For any column $c_x$ in the cycle (e.g., $c_1$), two of its elements are modified: $s'_{r_1,c_1}$ by $+\delta$ and $s'_{r_2,c_1}$ by $-\delta$. Thus, the sum $\sum_i s'_{i,c_1} = \sum_i s^{(0)}_{i,c_1} = C^{(0)}_1$. This holds for all columns $c_1, \ldots, c_L$. Columns not in the cycle are unaffected. Therefore, $C'_j = C^{(0)}_j$ for all $j$, and Objective 1 remains optimized.

- **Row Variances in $\mathcal{S}'$**: Consider row $r_1$. Two of its $k$ non-zero elements are now $1/k + \delta$ and $1/k - \delta$, while the other $k - 2$ remain $1/k$. The mean of non-zero elements in row $r_1$ is $\mu'_1 = \frac{1}{k}\left((k-2)\frac{1}{k} + (1/k + \delta) + (1/k - \delta)\right) = 1/k$. The variance of non-zero elements in row $r_1$ is:

$$Var_1(\mathcal{S}') = \frac{1}{k}\left[(k-2)\left(\frac{1}{k} - \frac{1}{k}\right)^2 + \left(\left(\frac{1}{k} + \delta\right) - \frac{1}{k}\right)^2 + \left(\left(\frac{1}{k} - \delta\right) - \frac{1}{k}\right)^2\right]$$

$$= \frac{1}{k}[0 + \delta^2 + (-\delta)^2] = \frac{2\delta^2}{k}$$

  Since $\delta > 0$ and $k \geq 2$, $Var_1(\mathcal{S}') > 0$. Similarly, for all rows $r_1, \ldots, r_L$ involved in the cycle, their variance of non-zero elements increases from $0$ to $2\delta^2/k$. Rows not in the cycle maintain zero variance. Thus, Objective 2 is achieved as the variance has increased for at least these $L$ rows.

### 4. Existence of the Desired State and Conclusion

The construction of $\mathcal{S}'$ from $\mathcal{S}^{(0)}$ demonstrates that if $k \geq 2$ and the support graph $G_{\mathcal{P}}$ (chosen to optimize Objective 1) contains a cycle, then a state $\mathcal{S}'$ exists satisfying the lemma's conditions. Objective 1 remains optimized, and Objective 2 is achieved because the variance of non-zero elements in rows participating in the cycle is strictly increased from zero.

Thus, under the stated assumption of cycle existence in an appropriately chosen support graph $G_{\mathcal{P}}$, the matrix $\mathcal{S}'$ is the desired state.

---

**Lemma 2** *For two sets of points $\mathcal{A}$ and $\mathcal{B}$ of equal size, it is always possible to partition $\mathcal{A} \cup \mathcal{B}$ such that $\mathcal{A} \cap \mathcal{B} = \varnothing$ and $|\mathcal{A}| = |\mathcal{B}|$.*

---

**proof C.2** *The lemma we aim to prove states: For two sets of points $\mathcal{A}$ and $\mathcal{B}$ of equal size, it is always possible to partition $\mathcal{A} \cup \mathcal{B}$ such that the components of this partition (also referred to as $\mathcal{A}$ and $\mathcal{B}$ in the conclusion of the lemma) satisfy $\mathcal{A} \cap \mathcal{B} = \varnothing$ and $|\mathcal{A}| = |\mathcal{B}|$.*

*For the lemma's assertion that this is "always possible" to hold, we interpret the conditions "$\mathcal{A} \cap \mathcal{B} = \varnothing$" and "$|\mathcal{A}| = |\mathcal{B}|$" in the conclusion as pertaining to the initially given sets $\mathcal{A}$ and $\mathcal{B}$ themselves. These sets must satisfy these conditions and thereby form the required partition of their union.*

*Let $\mathcal{A}$ and $\mathcal{B}$ be two sets of points. From the statement of the lemma and our interpretation, we establish the following premises:*

    *(i) The sets $\mathcal{A}$ and $\mathcal{B}$ are of equal size, i.e., there exists a non-negative integer $n$ such that $|\mathcal{A}| = |\mathcal{B}| = n$. (This is given by the lemma's hypothesis.)*

    *(ii) For $\mathcal{A}$ and $\mathcal{B}$ to serve as the components of the partition of $\mathcal{A} \cup \mathcal{B}$ and to satisfy the disjointness condition in the lemma's conclusion, we require that the sets $\mathcal{A}$ and $\mathcal{B}$ themselves are disjoint, i.e., $\mathcal{A} \cap \mathcal{B} = \varnothing$.*

*We now demonstrate that under premises (i) and (ii), the sets $\mathcal{A}$ and $\mathcal{B}$ form a partition of their union, $\mathcal{A} \cup \mathcal{B}$. First, consider the pair of sets $(\mathcal{A}, \mathcal{B})$ as a candidate partition for the set $U = \mathcal{A} \cup \mathcal{B}$. For $(\mathcal{A}, \mathcal{B})$ to be a valid partition of $U$, its components must satisfy two conditions:*

    • *The union of the components must be equal to the set being partitioned. Here, $\mathcal{A} \cup \mathcal{B}$ is by definition equal to $U$.*

    • *The components must be mutually disjoint. By premise (ii), we have $\mathcal{A} \cap \mathcal{B} = \varnothing$.*

*Thus, the sets $\mathcal{A}$ and $\mathcal{B}$ indeed form a valid partition of $\mathcal{A} \cup \mathcal{B}$.*

*Next, we verify that the components of this partition (i.e., $\mathcal{A}$ and $\mathcal{B}$) satisfy the specific properties mentioned in the conclusion of the lemma:*

    *1. The components of the partition are disjoint. This condition is $\mathcal{A} \cap \mathcal{B} = \varnothing$, which is true by premise (ii).*

    *2. The components of the partition are of equal size. This condition is $|\mathcal{A}| = |\mathcal{B}|$, which is true by premise (i).*

*Since the components of this partition (formed by $\mathcal{A}$ and $\mathcal{B}$ themselves) satisfy all the properties required by the lemma's conclusion, the lemma is proven.*

### C.4 Computational Overhead of $\mathbf{L_o}$

While $\mathbf{L_o}$ has quadratic complexity in theory, the actual overhead is negligible in practice due to the small number of activated experts ($k$) and efficient batched implementations. It does not present a bottleneck in our setup. Detailed experimental results are provided in Appendix J.

$\mathbf{L_o}$ involves pairwise projections among the $k$ selected expert outputs for each token, leading to a theoretical cost of $\mathcal{O}(N \cdot k^2 \cdot d)$. In practice, this cost remains manageable for three reasons.

**(1) Small $k$ in standard MoE practice.** Sparse MoE models typically keep $k$ small to control computation. In our experiments, we follow this convention, using configurations such as $k = 6$ in DeepSeek-V2-Lite. Given that $k \ll N$ and $k \ll d$, the quadratic factor contributes minimally to the overall training cost.

**(2) Efficient hardware execution.** The main operations in $\mathbf{L_o}$—inner products and pairwise projections—are highly parallelizable and efficiently implemented as batched matrix multiplications in frameworks such as PyTorch, running smoothly on modern GPUs.

**(3) Justified by empirical gains.** The modest increase in computation is offset by substantial and consistent performance improvements across diverse downstream tasks. This demonstrates that the regularization effect of $\mathbf{L_o}$ leads to meaningful gains in expert specialization without incurring prohibitive cost.

## D Datasets

**GSM8K** [12] is a benchmark designed to evaluate mathematical reasoning through 8,000 elementary and middle school word problems across arithmetic, algebra, geometry, and other topics. Each

problem comes with detailed step-by-step solutions, enabling models to learn chain-of-thought (CoT) reasoning strategies. The dataset is widely used to train and assess a model's ability to decompose multi-step questions logically and produce interpretable solutions.

**MATH500** [44] focuses on advanced mathematics with 500 university-level problems in calculus, linear algebra, abstract algebra, real analysis, and more. Problems typically require multi-step formal proofs, symbolic manipulation, and theoretical understanding, making it a strong test for mathematical maturity. Its emphasis on rigor and abstraction makes it ideal for developing specialized solvers and assessing formal reasoning depth.

**Numina** [41] is a large-scale math dataset containing approximately 24,000 problems ranging from primary to high school levels, annotated with explicit chain-of-thought reasoning steps. It is designed to teach models to perform structured, stepwise reasoning rather than shortcut memorization of solutions. The dataset is particularly effective for improving multi-step performance and explainability in math-based language models.

**MMLU** [31, 30] is a massive multitask benchmark with multiple-choice questions spanning 57 academic subjects, including science, humanities, law, and medicine. Each subject is stratified by difficulty (high school to expert level), allowing evaluation across a broad spectrum of general knowledge. MMLU is a widely adopted standard for testing cross-domain reasoning and factual recall in large language models (LLMs).

**MMLU-pro** [70] is an expert-level extension of MMLU that increases question difficulty by expanding answer choices and emphasizing multi-step, high-complexity problems. It targets challenging domains like STEM reasoning and policy analysis, where simple factual recall is insufficient. MMLU-pro is ideal for benchmarking models under professional-grade conditions with nuanced and layered reasoning requirements.

**BBH** [63] consists of hundreds of diverse tasks covering complex scenarios such as logical reasoning, language games, social sciences, and physical commonsense. Its design aims to challenge models on unconventional capabilities, such as counterfactual reasoning and cross-lingual transfer. Most BBH tasks are open-ended, requiring the integration of commonsense and creative thinking—for example, generating poetry or designing ethical AI frameworks.

**GLUE** [66, 71, 61, 18, 1, 77, 13, 23, 40] is a foundational NLP benchmark combining nine language understanding tasks such as sentiment classification, sentence similarity, and entailment detection. It provides a standardized framework to assess general-purpose language comprehension and model transferability across tasks. GLUE has been instrumental in shaping the early progress and comparison of pre-trained language models.

**HumanEval** [10] is a code generation benchmark released by OpenAI, containing 164 Python programming tasks with unit test specifications. It focuses on assessing a model's ability to synthesize functionally correct, efficient, and stylistically appropriate code from natural language prompts. HumanEval remains a key benchmark for evaluating reasoning, planning, and syntax correctness in code generation models.

**MBPP** [4] features thousands of Python programming problems based on real-world development scenarios like string parsing, API use, and algorithm design. Each task includes input/output specifications and test cases, enabling automated evaluation of code correctness and performance. MBPP is widely used to train and evaluate models for practical software engineering and step-by-step code synthesis.

**LiveBench** [76] is a real-time evaluation benchmark capturing dynamic user-model interactions from deployment environments like chatbots or decision engines. It tracks response latency, robustness, and contextual consistency in streaming or multi-turn settings. LiveBench is designed to reveal edge-case failures and test a model's adaptability under realistic, time-sensitive constraints.

**GPQA** [59] is a high-difficulty multiple-choice dataset written by domain experts in biology, physics, and chemistry, targeting scientific reasoning at an expert level. Questions often require interdisciplinary integration and reasoning across theory, data interpretation, and experimental design. GPQA is ideal for probing a model's capabilities in abstract scientific synthesis and expert-level domain understanding.

# E   Metrics

**MaxVio$_{\text{global}}$** [68] is a metric introduced to quantify load imbalance in Mixture-of-Experts (MoE) models. A lower value indicates more balanced expert utilization, while a higher value reflects severe imbalance. It evaluates global load balance across the entire validation set, reflecting long-term efficiency and fairness in expert usage.

$$MaxVio_{\text{global}} = \frac{max_i Load_i - \overline{Load_i}}{\overline{Load_i}} \tag{36}$$

where:

- $Load_i$ is the number of tokens assigned to expert $i$.
- $\overline{Load_i}$ is the average (ideal balanced) load across experts.

**Accuracy (ACC)** is a metric that measures the proportion of correct predictions made by a model. It's calculated as the number of correct predictions divided by the total number of predictions.

$$ACC = \frac{\text{Number of Correct Predictions}}{\text{Total Number of Predictions}} \tag{37}$$

**Silhouette Coefficient** is a metric used to evaluate the quality of clustering. It measures how similar a data point is to its own cluster compared to other clusters, considering both cohesion and separation. Values range from -1 to +1, where a higher value indicates that the object is well-matched to its own cluster and poorly matched to neighboring clusters.

$$s(i) = \frac{b(i) - a(i)}{\max\{a(i), b(i)\}} \tag{38}$$

where:

- $a(i)$ is the average distance from sample $i$ to all other points in the same cluster (intra-cluster dissimilarity).
- $b(i)$ is the minimum average distance from sample $i$ to all points in any other cluster (inter-cluster dissimilarity).

**Expert Overlap** primarily describes a feature in Mixture of Experts (MoE) models where specialized subnetworks (experts) are not entirely distinct. These experts might share parameters or have intentionally intersecting knowledge domains to process similar types of data or tasks.

The actual number of neighbors, $k'$, used for an input parameter $k_{param}$ and $N$ total embeddings is:

$$k' = \min(k_{param}, N - 1) \tag{39}$$

The overlap score for an individual embedding $e_i$, denoted $O_i$, is:

$$O_i = \frac{1}{k'} \sum_{e_j \in N_i(k')} \mathbb{I}(l_j \neq l_i) \tag{40}$$

The overall expert overlap score, $S_{overlap}$, is the average of these individual scores:

$$S_{overlap} = \frac{1}{N} \sum_{i=1}^{N} O_i = \frac{1}{N} \sum_{i=1}^{N} \left( \frac{1}{k'} \sum_{e_j \in N_i(k')} \mathbb{I}(l_j \neq l_i) \right) \tag{41}$$

where:

- $N$ is the total number of embeddings.
- $k_{param}$ is the user-specified number of nearest neighbors.
- $k'$ is the adjusted number of nearest neighbors, $\min(k_{param}, N - 1)$, used in the calculation (meaningful for $k' > 0$).
- $e_i$ represents the $i$-th embedding from the set of embeddings $E = \{e_1, e_2, \ldots, e_N\}$.

- $l_i$ is the expert label corresponding to the embedding $e_i$, from the set of labels $L = \{l_1, l_2, \ldots, l_N\}$.
- $N_i(k')$ is the set of $k'$ nearest neighbors of embedding $e_i$, excluding $e_i$ itself.
- $\mathbb{I}(\cdot)$ is the indicator function, which is 1 if the condition (e.g., $l_j \neq l_i$) is true, and 0 otherwise.

The $S_{overlap}$ score ranges from 0 to 1. A score of 0 indicates no overlap (all $k'$ nearest neighbors of any point share its label), while a score of 1 indicates complete overlap (all $k'$ nearest neighbors of any point have different labels). A lower score generally signifies better expert separation in the embedding space.

**Routing Variance** refers to the inconsistency or fluctuation in how the gating network distributes inputs to different expert sub-models.It measures the variability in which expert(s) are chosen for similar inputs or over time, reflecting the stability of the routing decisions.

$$RoutingVariance = \frac{1}{N_E} \sum_{j=1}^{N_E} \left( \left( \frac{1}{N_S} \sum_{i=1}^{N_S} g_j(x_i) \right) - \frac{1}{N_E} \right)^2 \tag{42}$$

where:

- $N_E$: Total number of experts.
- $N_S$: Number of input samples.
- $g_j(x_i)$: Gating probability of input $x_i$ being assigned to expert $j$.

**Root Mean Square Error (RMSE)** is a standard statistical metric used to evaluate the performance of a model by quantifying the magnitude of error between predicted and observed values. Lower RMSE values signify a closer fit of the model to the data, indicating higher predictive accuracy.

$$RMSE = \sqrt{\frac{1}{n} \sum_{i=1}^{n} (y_i - \hat{y}_i)^2} \tag{43}$$

where:

- $n$ is the total number of observations.
- $y_i$ represents the $i$-th actual (observed) value.
- $haty_i$ represents the $i$-th predicted value.
- sum denotes the summation over all observations from $i = 1$ to n.

# F Implementation Details

**DeepSeek-Moe-16B**[14] `DeepSeekMoE-16B` is a Mixture-of-Experts (MoE) language model with 16.4B parameters. It employs an innovative MoE architecture, which involves two principal strategies: fine-grained expert segmentation and shared experts isolation. It is trained from scratch on 2T English and Chinese tokens, and exhibits comparable performance with `DeekSeek 7B` and `LLaMA2-7B`, with only about 40% of computations.

**Moonlight-16B-A3B**[48] `Moonlight-16B-A3B` is a 16 billion-parameter Mixture-of-Experts (MoE) language model developed by Moonshot AI. It employs the Muon optimizer to train on 5.7 trillion tokens, achieving a new Pareto frontier of performance per FLOP. Available in both a 3 billion activated-parameter inference configuration and the full 16 billion-parameter scale, it outperforms comparable models such as `Llama3-3B` and `Deepseek-v2-Lite` while requiring significantly less compute. The model and its instruction-tuned variant are open-source on Hugging Face, with checkpoints and a memory- and communication-efficient Muon implementation provided to foster further research.

**DeepSeek-V2-Lite**[16] `DeepSeek-V2` is a strong Mixture-of-Experts (MoE) language model characterized by economical training and efficient inference. `DeepSeek-V2` adopts innovative

architectures including Multi-head Latent Attention (MLA) and DeepSeekMoE. MLA guarantees efficient inference through significantly compressing the Key-Value (KV) cache into a latent vector, while DeepSeekMoE enables training strong models at an economical cost through sparse computation.

We integrate our balance loss $\mathcal{L}_{\text{balance}}$ into each MoE layer by modifying the model's modeling file. During training, due to device computational resource constraints, we employ LoRA for fine-tuning (note that the sole difference from full-parameter fine-tuning lies in the smaller number of parameters, with no fundamental difference in the training mechanism). LoRA uses standard configurations (rank 32, LoRA $\alpha = 128$, learning rate $1 \times 10^{-4}$, batch size 32, dropout 0.1). We keep several original training hyperparameters in the model configuration, including the weight $\alpha$ of $\mathcal{L}_{\text{aux}}$ defaulting to 0.001. Settings like top-$k$ activation and routing scoring function type match each model's default configurations. The weights $\beta$ and $\gamma$ for our $\mathcal{L}_o$ and $\mathcal{L}_v$ are set identically to $\alpha$. During inference, we first merge the trained LoRA into the base model, then infer using vllm with gpu_memory_utilization $= 0.9$. Evaluation uses three validation methods: rule-based extraction, GPT-4o, and human experts.

## G  Baselines

**GShard**[39] GShard is a pioneering Mixture-of-Experts (MoE) architecture developed by Google Research, designed for massively parallelized training across thousands of devices. It introduces automatic tensor sharding to scale model parameters and data efficiently, achieving dynamic load balancing during distributed computation. Trained on 600 billion tokens, GShard demonstrated breakthrough performance in multilingual machine translation across 100+ languages while maintaining linear computational cost scaling. Its innovations in sparse expert routing and memory optimization laid the foundation for subsequent large-scale MoE systems.

**ST-MoE**[85] ST-MoE (Sparsely-Trained Mixture-of-Experts) is a compute-efficient framework from Google that enhances MoE model stability through specialized training techniques. It employs a novel router design with expert dropout and auxiliary loss terms to prevent mode collapse during sparse activation. Scaling to 269 billion parameters with only 3 billion active parameters per token, ST-MoE achieves state-of-the-art results on language modeling and reasoning tasks while using 5-7x less compute than dense counterparts. The architecture incorporates parameter-sharing strategies across experts to improve sample efficiency and reduce memory footprint.

**Loss-Free Balancing**[68] Loss-Free Balancing addresses the routing imbalance in MoE models without explicit optimization objectives. Traditional approaches rely on auxiliary loss functions to enforce expert load balancing, often at the cost of model performance or computational efficiency. This method dynamically adjusts entropy constraints on routing decisions and incorporates an adaptive activation threshold mechanism for sparse gating, achieving balanced expert utilization without auxiliary losses. It preserves primary task performance while demonstrating robustness in large-scale multi-task scenarios.

**With Aux Loss**[46] This classical load-balancing strategy for MoE training introduces explicit auxiliary losses during routing to constrain variance in expert utilization. Two complementary designs are implemented: (1) soft regularization terms (e.g., L2 penalties) based on expert selection frequency, and (2) probability redistribution strategies for cold-start experts. While effective in mitigating long-tail distribution issues, it requires careful tuning of loss weights to avoid interference with the primary task.

## H  Experiments Details

### H.1  Hyperparameter Sensitivity

To address the importance of hyperparameter sensitivity, we conducted experiments varying the values of the loss weights $\alpha$ (for $L_{aux}$), $\beta$ (for $L_o$), and $\gamma$ (for $L_v$) across different magnitudes.

For reference, our overall loss function $L$ is defined as the sum of $L_h$ and $L_{balance}$. The balance loss $L_{balance}$ is defined as:

$$L = L_h + L_{balance} \tag{44}$$
$$L_{balance} = \alpha \cdot L_{aux} + \beta \cdot L_o + \gamma \cdot L_v \tag{45}$$

It is worth noting that we apply a dynamic balancing mechanism to ensure fair weighting across different loss terms. Specifically, because the orthogonality and variance losses ($L_o$ and $L_v$) may have different initial scales, we first normalize them using dynamic scaling factors. This brings their magnitudes roughly in line with the auxiliary loss $L_{aux}$. Only after this normalization do we apply the hyperparameters $\alpha, \beta$, and $\gamma$ to control their contributions to the total loss.

The table below summarizes our results across several representative benchmarks under four different settings (DS v2 lite), as shown in Table 3.

Table 3: Hyperparameter sensitivity analysis. We evaluate performance across multiple benchmarks with different combinations of $\alpha, \beta, \gamma$ (DS v2 lite).

| $\alpha, \beta, \gamma$ | MMLU | GPQA | HumanEval | GSM8K | MATH500 | MaxVioGlobal |
|---|---|---|---|---|---|---|
| $10^{-3}, 10^{-3}, 10^{-3}$ | 35.59 | 28.76 | 43.58 | 50.94 | 49.33 | 2.52 |
| $10^{-3}, 10^{-4}, 10^{-3}$ | 31.24 | 25.52 | 41.62 | 46.63 | 46.23 | 3.05 |
| $10^{-3}, 10^{-3}, 10^{-4}$ | 33.52 | 27.35 | 39.52 | 48.30 | 49.09 | 2.77 |
| $10^{-4}, 10^{-3}, 10^{-3}$ | 30.74 | 26.90 | 42.85 | 49.62 | 44.54 | 4.57 |

From the results in Table 3, we observe that the setting where $\alpha = \beta = \gamma = 10^{-3}$ consistently yields the best performance across tasks. This suggests that the performance is optimal when all three loss weights $\alpha, \beta$, and $\gamma$ are set to the same value.

Furthermore, our method demonstrates strong robustness across different hyperparameter magnitudes. When any of the coefficients is varied within one order of magnitude ($\pm 1$), i.e., $10^{-3}$ vs $10^{-4}$, the results remain stable and close to optimal. This indicates that our method is not overly sensitive to these hyperparameters and can be considered robust in practical applications.

## H.2 Configurations and Base Model Performance

A discrepancy between our reported results and the original model figures from public citations (e.g., Moonlight, DeepSeek) was observed. This disparity primarily arises from differences in model versions, prompting strategies, and inference settings. We clarify these differences below:

- **Model versions:** The public figures are typically based on instruction-tuned models. In contrast, our work starts from their pretrained base versions, which have no preference or SFT (Supervised Fine-Tuning) data, leading to inherently different performance baselines.

- **Prompting strategies:** Our evaluation is conducted in a **zero-shot** setting without hand-crafted few-shot prompts or demonstrations, which are often used in official evaluations.

- **Inference length:** We uniformly limit the generation to **512 max new tokens** due to computational constraints. In contrast, official results often use 8k–32k tokens, which notably benefits reasoning-heavy tasks like MMLU and HumanEval.

To quantify this impact, we evaluated both base and our fine-tuned models under the same, matched token budget (512 tokens). The results are summarized in Table 4. We also analyze the effect of increasing the token length for the Kimi model in Table 5.

Table 4: Performance Comparison under Matched Inference Settings (512 max new tokens).

| Method | MMLU | GPQA | HumanEval | GSM8K | MATH500 | MaxVioGlobal |
|---|---|---|---|---|---|---|
| Base (ds) | 28.46 | 22.45 | 42.64 | 28.76 | 7.34 | 4.63 |
| Ours (ds) | 33.35 | 25.15 | 63.30 | 35.00 | 10.82 | 2.19 |
| Base (ds-v2) | 26.56 | 20.33 | 31.34 | 22.57 | 15.69 | 6.97 |
| Ours (ds-v2) | 35.59 | 28.76 | 43.58 | 50.94 | 49.33 | 2.52 |
| Base (Kimi) | 34.23 | 28.33 | 55.67 | 81.23 | 53.76 | 8.37 |
| Ours (Kimi) | 40.36 | 32.01 | 70.64 | 87.62 | 59.64 | 7.23 |

Table 5: Effect of Increasing Max New Tokens (Kimi).

| Method | MMLU | GPQA | HumanEval | GSM8K | MATH500 | MaxVioGlobal |
|--------|------|------|-----------|-------|---------|--------------|
| Base 512 | 34.23 | 28.33 | 55.67 | 81.23 | 53.76 | 8.37 |
| Base 1024 | 37.74 | 31.42 | 60.24 | 82.42 | 55.62 | 8.22 |
| Base 2048 | 45.43 | 33.52 | 62.09 | 82.73 | 60.13 | 9.01 |
| Ours 512 | 40.36 | 32.01 | 70.64 | 87.62 | 59.64 | 7.23 |
| Ours 1024 | 45.63 | 35.22 | 73.23 | 88.31 | 65.23 | 7.14 |
| Ours 2048 | 47.95 | 36.78 | 73.62 | 86.25 | 69.82 | 6.87 |

As shown in Table 4, under identical inference constraints (512 tokens), our method consistently outperforms the original base models. Furthermore, Table 5 demonstrates that our method retains its leading performance even when the generation length is extended. We note that due to computational resource constraints, we were unable to reproduce the official results from other papers, which often utilize significantly longer sequence lengths (e.g., 8k-32k tokens).

### H.3 Performance Under Larger and More Diverse Training Data

We conducted an experiment to evaluate the impact of training data size and diversity on the effectiveness of our method.

#### H.3.1 Motivation from Single-Task Settings

As noted in the introduction, our method is motivated by the observation that in post-training scenarios, the training data is often domain-specific and less diverse. This results in highly skewed token distributions, which intensifies the conflict between load balancing (which encourages even token-to-expert allocation) and expert specialization (which encourages domain-specific token routing). Our method was designed to explicitly address this tension.

#### H.3.2 Performance on Mixed and Richer Datasets

To test whether our method still performs well with more diverse training data, we constructed a mixed dataset combining Numina (math), GPQA (science), and HumanEval (coding), totaling 18k examples. We fine-tuned the Moonlight (Kimi) model for 3 epochs on this combined dataset. The results are summarized in Table 6.

Table 6: Performance comparison on a larger, mixed dataset (Numina, GPQA, HumanEval) using the Moonlight (Kimi) model.

| Method | MMLU | GPQA | HumanEval | GSM8K | MATH500 | MaxVioGlobal |
|--------|------|------|-----------|-------|---------|--------------|
| Base | 34.23 | 28.33 | 55.67 | 81.23 | 53.76 | 8.37 |
| AuxOnly | 36.98 | 31.34 | 67.53 | 84.83 | 62.29 | 7.07 |
| AuxFree | 35.87 | 29.48 | 68.83 | 86.29 | 63.84 | 7.28 |
| **Ours** | **45.38** | **37.01** | **78.93** | **92.92** | **67.83** | **7.11** |

As shown, our method continues to outperform all baselines, even when trained on a significantly larger and more diverse dataset. This demonstrates that our approach remains robust and effective beyond constrained single-task settings.

## I  More Baselines and MoE Architectures

### I.1 Comparison with Additional Baselines

To provide a more comprehensive evaluation, we expanded our set of comparison methods to include two additional state-of-the-art baselines. We re-evaluated all methods on the most comprehensive subsets of our benchmark suite.

The added baselines are:

- **Dynamic Routing MoE (ERNIE 4.5) [5]**: This is a strong recent baseline that introduces a multimodal, heterogeneous MoE architecture. It supports both parameter sharing across modalities and modality-specific expert specialization. The ERNIE 4.5 family includes multiple model scales (e.g., 47B and 3B active parameters) and has shown competitive performance on various text and multimodal benchmarks.
- **SIMBAL (Similarity-Preserving Routers) [55]**: This is a recent method addressing expert load balancing in sparse MoE models. Instead of enforcing uniform routing via conventional load balancing loss, SIMBAL introduces an orthogonality-based regularization. This aligns the router's Gram matrix with the identity matrix, encouraging similar input tokens to be routed to similar experts, thereby reducing redundancy and improving consistency in expert utilization.

A summary of the key results is presented in Table 7.

Table 7: Performance comparison against additional state-of-the-art baselines. Our method demonstrates superior performance and achieves the best (lowest) load balance score (MaxVioGlobal).

| Method | MMLU | GPQA | HumanEval | GSM8K | MATH500 | MaxVioGlobal |
|---|---|---|---|---|---|---|
| ERNIE 4.5 LBL | 32.44 | 27.45 | 37.32 | 47.24 | 42.63 | 3.45 |
| SIMBAL | 31.89 | 27.64 | 39.45 | 48.75 | 45.36 | 4.56 |
| **Ours** | **35.59** | **28.76** | **43.58** | **50.94** | **49.33** | **2.52** |

As shown in Table 7, after incorporating these two additional state-of-the-art baselines, our approach continues to deliver the best overall performance. Across all six representative tasks, our method either matches or surpasses the strongest new baseline, while simultaneously maintaining the lowest MaxVioGlobal (indicating better load balance). These additional results confirm that the improvements reported in the main paper are not an artifact of the original baseline selection but hold against the latest alternatives as well.

## I.2 Performance on Diverse MoE Architectures

To further validate the generality of our method, we extended our evaluation to more diverse MoE architectures. Our initial experiments focused on DeepSeek and Moonlight models due to their strong open-source performance and recent community adoption. To broaden this scope, we additionally evaluated our method on two structurally different models: Mixtral and Phi-MoE, which adopt distinct routing strategies and omit shared experts.

The results, shown in Table 8, demonstrate that our method continues to outperform baselines across all tasks on these diverse architectures.

Table 8: Performance comparison on diverse MoE architectures (Mixtral and Phi-MoE).

| Method | MMLU | GPQA | HumanEval | GSM8K | MATH500 | MaxVioGlobal |
|---|---|---|---|---|---|---|
| *Mixtral Architecture* | | | | | | |
| Mixtral-Base | 43.32 | 15.34 | 33.23 | 52.42 | 20.63 | 6.32 |
| Mixtral-AuxOnly | 50.56 | 18.84 | 39.74 | 58.73 | 28.74 | 3.25 |
| Mixtral-AuxFree | 49.73 | 20.14 | 36.06 | 56.96 | 29.84 | 3.58 |
| **Mixtral-Ours** | **52.63** | **20.74** | **37.73** | **61.74** | **33.42** | **3.54** |
| *Phi-MoE Architecture* | | | | | | |
| PhiMoE-Base | 51.73 | 34.52 | 66.46 | 84.52 | 41.84 | 7.53 |
| PhiMoE-AuxOnly | 57.24 | 34.21 | 71.32 | 85.21 | 42.94 | 5.21 |
| PhiMoE-AuxFree | 53.52 | 35.32 | 70.45 | 86.24 | 44.52 | 5.35 |
| **PhiMoE-Ours** | **59.63** | **35.87** | **76.23** | **88.32** | **44.79** | **5.32** |

These results further demonstrate the generality of our approach, showing its effectiveness across MoE models with different underlying architectural designs.

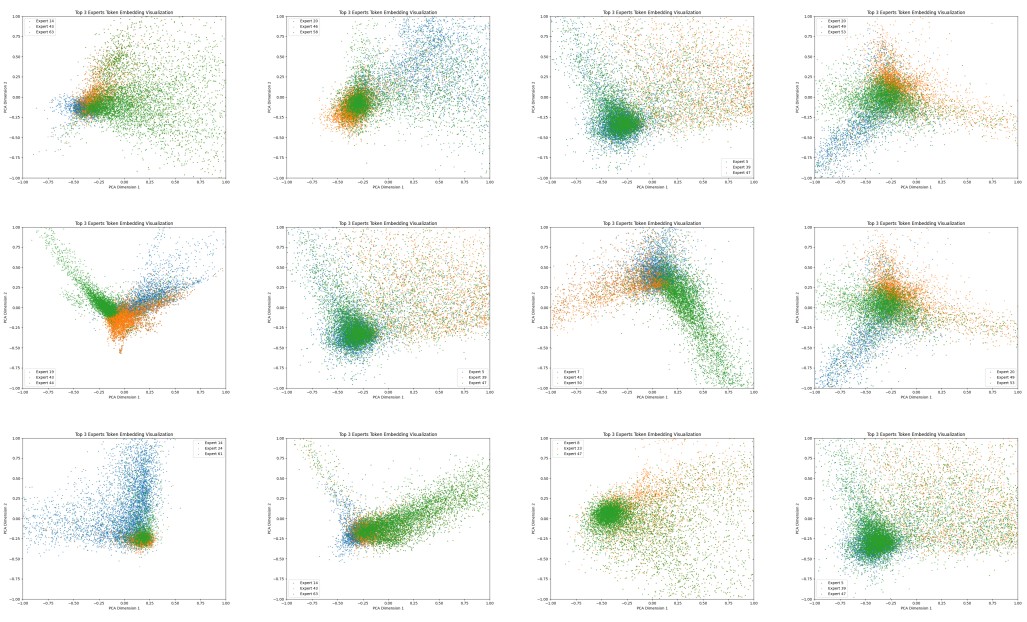

Figure 5: Selected Images (4×3)

## J  Training Overhead

While our method introduces some additional computation due to the proposed regularization losses, the training time remains within a practical range and compares favorably with existing baselines. We report the average step time (in seconds per iteration) on the DeepSeek V2 Lite model using a batch size of 32. The results are summarized in Table 9.

Table 9: Training time comparison (seconds per iteration) on the DeepSeek V2 Lite model (batch size 32).

| Method | Time (s/iter) |
| --- | --- |
| Ours | 11.5 |
| Only Aux | 10.7 |
| Aux Free | 9.8 |
| GShard | 14.8 |
| ST-MoE | 12.1 |

Our approach incurs moderate overhead compared to load-balancing-only methods like "Aux Free" and "Only Aux," but remains significantly more efficient than GShard and ST-MoE. Given that our method achieves up to a **23.79% performance improvement** across benchmarks (as reported in the abstract), we believe this efficiency-performance trade-off is well justified.

## K  Visualization

Figures 5 present the PCA projection of token embeddings assigned to the top 3 most active experts from baseline models. The significant overlap among different colors suggests that the token representations routed to different experts are not well separated. This indicates high expert overlap and a lack of clear specialization among experts in the representation space.

