# OpenReview forum: "Advancing Expert Specialization for Better MoE"
_NeurIPS.cc/2025/Conference — NeurIPS 2025 oral_

### Official Review · Reviewer_Ypny · 2025-06-29

**Clarity:** 3
**Significance:** 2
**Originality:** 2
**Rating:** 4
**Confidence:** 4

**Summary:**

In this paper, the authors address the challenge of poor expert specialization in standard MoE models trained with load-balancing loss. The authors argue that the conventional load-balancing auxiliary loss leads to expert overlap or uniform routing. The paper proposes to mitigate that using two additional loss functions: 1) an orthogonality loss to encourage experts to learn unique functions and 2) a variance loss to promote more discriminative routing. The newly added loss functions are used for finetuning three pre-existing MoE models without any architecture modifications. Results are presented for 11 benchmarks including language, reasoning, science, math, and code evaluation.

**Questions:**

See the section above.

Minor point:
Line 66: Two periods at the end of the sentence.

**Ethical Concerns:**

["NO or VERY MINOR ethics concerns only"]

**Final Justification:**

The authors have addressed several of my concerns, including adding information on training settings, reporting overhead, and reporting downstream task performance.

**Limitations:**

The authors do not sufficiently address the limitations of their work, including discussion of the computational overhead for the added loss functions and a sensitivity analysis for MoE training with 4 loss functions.

**Quality:**

2

**Strengths And Weaknesses:**

**Strengths:**
- The paper is well-written and addresses a significant, well-known conflict in MoE training.
- The results show the model outperforms alternative regularization methods for the finetuned models.
- The ablation showing the impact of different loss functions on the behavioural states of the models is very informative.

**Weaknesses:**
- **Missing pre-finetuning baseline results**: The authors do not report the performance of the original, pre-finetuning MoE models. Based on public figures for models like Moonlight and DeepSeek, the finetuning process described in the paper appears to significantly degrade performance on key benchmarks (e.g., MMLU, BBH). For example, the original accuracy of Moonlight MoE model on MMLU, MMLU pro, MBPP, and BBH are 70, 42.4, 63.8, 65.2, which are above the best reported numbers in Table 1. Similar behaviour is observed for DeepSeek models: MMLU 58.3 compared to the 35.59 of the proposed method, or BBH 44.1 compared to 38.84.

- **LoRA mitigating performance loss**: The authors focus exclusively on LoRA finetuning of high-quality, pre-converged models, but this crucial detail is confined to the appendix. Given the high performance of the original base models, it should be clarified whether their method is simply mitigating performance loss during finetuning rather than providing a net gain over the original models.

- **Complexity of multi-objective training with 4 loss functions:** While the authors only focus on finetuning already converged high-quality MoE models, it is unclear how the complex optimization of four separate loss functions behaves when training MoEs from scratch or when upcycling dense models, which are more challenging and common scenarios. The paper also lacks a sensitivity analysis for the loss coefficients (`α, β, γ`), making reproducibility and application difficult.

- **Computational Overhead:** The paper does not discuss the computational overhead of the proposed loss functions. In particular, the orthogonality loss, which requires pairwise comparisons of expert outputs for the top-k selections, appears prohibitively expensive, especially with the top-6 expert configuration used in the experiments. A discussion of the added training cost and memory is essential.

- **Limited MoE architectures:** All three models evaluated (DeepSeek, DeepSeek-Lite, Moonlight) share an identical MoE structure (64 experts, top-6 selection, 2 shared experts). To strengthen their claims, the authors should test their method on more diverse architectures like Mixtral and Phi-MoE, which use different routing strategies and do not include shared experts.

---

> ### Author Rebuttal · Authors · 2025-07-31
>
> **Dear Reviewer Ypny,**
>
> We sincerely thank the reviewer for the thoughtful and detailed feedback. The comments raise several key issues, including:
>
> 1. The lack of pre-finetuning baselines and potential performance drop;
>
> 2. Whether the method mitigates LoRA-induced degradation;
>
> 3. The complexity and reproducibility of multi-loss optimization;
>
> 4. Missing analysis of computational overhead;
>
> 5. Limited architecture diversity;
>
> 6. Insufficient discussion of limitations.
>
> We appreciate these valuable insights and address each point below.
>
> ---
>
> **1. Missing pre-finetuning baseline results and apparent performance drop**
>
> The discrepancy between our reported results and the original model figures primarily arises from differences in model versions, prompting strategies, and inference settings.
>
> - **Model versions**: The public figures cited (e.g., Moonlight, DeepSeek) are based on *instruction-tuned* models, while we start from their *pretrained base versions* with no preference or SFT data, leading to inherently different performance baselines.
> - **Prompting strategies**: Our evaluation is conducted in a **zero-shot** setting without handcrafted few-shot prompts or demonstrations, which are often used in official evaluations.
> - **Inference length**: We uniformly limit the generation to **512 max new tokens** due to computational constraints. In contrast, official results use **8k–32k** tokens, which notably benefits reasoning-heavy tasks like MMLU and HumanEval. To quantify this impact, we evaluated both base and our fine-tuned models under the same token budget. Results are summarized below:
>
> *Performance Comparison under Matched Inference Settings*
>
> |Method     |MMLU |GPQA |HumanEval|GSM8K|Math500|MaxvioGlobal|
> |---|---|---|---|---|---|---|
> |Base (ds) |28.46|22.45|42.64|28.76|7.34 |4.63|
> |Ours (ds) |33.35|25.15|63.30|35.00|10.82|2.19|
> |Base (ds-v2) |26.56|20.33|31.34|22.57|15.69|6.97|
> |Ours (ds-v2) |35.59|28.76|43.58|50.94|49.33|2.52|
> |Base (Kimi) |34.23|28.33|55.67|81.23|53.76|8.37|
> |Ours (Kimi) |40.36|32.01|70.64|87.62|59.64|7.23|
>
>
>
> *Effect of Increasing Max New Tokens (Kimi)*
>
> |Method    |MMLU |GPQA |HumanEval|GSM8K|Math500|MaxvioGlobal|
> |---|---|---|---|---|---|---|
> |Base 512 |34.23|28.33|55.67|81.23|53.76|8.37|
> |Base 1024|37.74|31.42|60.24|82.42|55.62|8.22|
> |Base 2048|45.43|33.52|62.09|82.73|60.13|9.01|
> |Ours 512 |40.36|32.01|70.64|87.62|59.64|7.23|
> |Ours 1024|45.63|35.22|73.23|88.31|65.23|7.14|
> |Ours 2048|47.95|36.78|73.62|86.25|69.82|6.87|
>
> As shown, under identical inference constraints, our method outperforms the original models, and retains leading performance even when length is extended. We will include these clarifications and additional results in the revised manuscript. Due to computational resource constraints, we were unable to reproduce the official results with longer sequence lengths, and we sincerely apologize for this limitation.
>
> ---
>
> **2. Clarifying Performance Gains with Our Method**
>
> We would like to clarify that our approach provides **a net performance gain**, not just recovery from potential LoRA-induced loss.
>
> First, from a theoretical perspective, the proposed losses $\mathcal{L}_o$ and $\mathcal{L}_v$ are designed to actively enhance expert specialization while preserving load balance (as supported by Lemmas 1 and 2), which defines a constructive optimization objective beyond damage control (Detail in Reviewer 9cq6 Q1 ).
>
> Second, as shown, when evaluated under consistent inference settings (e.g., token limit = 512), our fine-tuned models outperform the original base models across multiple benchmarks. These results confirm that our method achieves genuine improvement rather than merely compensating for LoRA finetuning limitations.
>
> ---
>
> **3. On applicability to pre-training and loss coefficient sensitivity**
>
> We appreciate the reviewer’s insightful suggestions. While our work focuses on post-training due to limited compute and data resources, the proposed losses are theoretically applicable to pre-training. In fact, token distributions in pre-training are more balanced, making it easier to satisfy both load balancing and specialization objectives. We will include this as a future direction in Section 6 (Detail of post-training in Reviewer 9cq6  Q2).
>
> For hyperparameter sensitivity, we conducted experiments varying α, β, and γ, and adopted a dynamic scaling mechanism to normalize the magnitudes of $\mathcal{L}_o$ and $\mathcal{L}_v$ relative to $\mathcal{L}_{aux}$. This ensures stable optimization across losses. Results show our method remains robust under a wide range of settings.
>
> |$\alpha,\beta,\gamma$|MMLU|GPQA|HumanEval|GSM8K|MATH500|MaxVioGlobal|
> |---|---|---|---|---|---|---|
> |1e-3,1e-3,1e-3|35.59|28.76|43.58|50.94|49.33|2.52|
> |1e-3,1e-4,1e-3|31.24|25.52|41.62|46.63|46.23|3.05|
> |1e-3,1e-3,1e-4|33.52|27.35|39.52|48.30|49.09|2.77|
> |1e-4,1e-3,1e-3|30.74|26.90|42.85|49.62|44.54|4.57|
>
>
> The default setting achieves the best trade-off between general performance (MMLU) and specialization effectiveness (MaxVioGlobal), demonstrating robustness to coefficient variation.
>
> ---
>
> **4. On computational overhead**
>
> While the orthogonality loss ($\mathcal{L}_o$) has a theoretical complexity of $O(N \cdot k^2 \cdot d)$, this overhead is practically manageable. In standard MoE setups, the number of active experts $k$ is a small constant (e.g., 6 in DeepSeek models), with $k \ll N$ and $k \ll d$. Moreover, the required pairwise projections are highly optimized on modern GPU hardware via batched matrix operations. Therefore, $\mathcal{L}_o$ introduces limited additional cost and does not become a training bottleneck in practice.
>
> ---
>
> **5. On the diversity of evaluated MoE architectures**
>
> We acknowledge the reviewer’s suggestion to validate our method on more diverse MoE architectures. Our initial experiments focused on DeepSeek and Moonlight models due to their strong open-source performance and recent community adoption, making them credible testbeds for practical effectiveness.
>
> In response to this concern, we have additionally evaluated our method on two structurally different models: Mixtral and Phi-MoE, which adopt distinct routing strategies and omit shared experts. As shown below, our method continues to outperform baselines across all tasks.
>
> |Method|MMLU|GPQA|HumanEval|GSM8K|Math500|MaxvioGlobal|
> |---|---|---|---|---|---|---|
> |MixtralBasemodel|43.32|15.34|33.23|52.42|20.63|6.32|
> |MixtralAuxOnly|50.56|18.84|39.74|58.73|28.74|3.25|
> |MixtralAuxFree|49.73|20.14|36.06|56.96|29.84|3.58|
> |MixtralOurs|52.63|20.74|37.73|61.74|33.42|3.54|
> |PhiMoEBasemodel|51.73|34.52|66.46|84.52|41.84|7.53|
> |PhiMoEAuxOnly|57.24|34.21|71.32|85.21|42.94|5.21|
> |PhiMoEAuxFree|53.52|35.32|70.45|86.24|44.52|5.35|
> |PhiMoEOurs|59.63|35.87|76.23|88.32|44.79|5.32|
>
> We will include these results in the revised version to better demonstrate the generality of our approach.
>
> ---
>
> **6. On writing issues and limitations discussion**
>
> We thank the reviewer for the careful reading.
>
> - We will correct the punctuation issue on line 66 in the revised version.
>
> - We will rewrite and expand Section 6 into a more comprehensive *Limitations and Future Work* discussion to improve clarity and completeness.
>
> ---
>
> We sincerely thank the reviewer again for raising many important and constructive concerns. While this review included the most critical questions, we believe we have now addressed them thoroughly in our rebuttal, including the pre-finetuning comparisons, training setup clarification, loss function sensitivity, and generalization to new architectures.
>
> We would like to reiterate that our paper proposes a novel and effective framework to jointly improve expert specialization and load balancing in MoE models, supported by theoretical guarantees, carefully designed regularization losses, and strong empirical results across 11 benchmarks. The remaining three reviewers have explicitly acknowledged the clarity, novelty, and significance of our contributions.
>
> We hope our clarifications have addressed your concerns. If you find the revised explanations and results satisfactory, we would greatly appreciate your consideration in recommending the paper for further discussion.

---

> > ### Comment · Reviewer_Ypny · 2025-08-05
> >
> > I would like to thank the authors for their significant effort to improve the clarity of their method and evaluations. To further strengthen the paper, could you please report the relative increase in training time due to the added loss functions in the paper? I appreciate the added experiments on Mixtral and Phi-MoE.
> >
> > On a separate note, it is concerning that the authors chose not to provide important training details (now provided in their response to reviewer 9cq6) in the original submission neither in the main body nor in the appendix (e.g. amount of training data - 6k, extended data curation, number of epochs, small max length - 512, etc.). Including such details is crucial for assessment of the method.
> >
> > This post training setting explained in response to **9cq6** is quite constrained. In this setting, the authors always train and evaluate on the same domain and do not evaluate the generalization of the trained model on other downstream task. A key concern is whether improving `in-domain` performance comes at the cost of performance degradation on other `downstream tasks`? The trade-off between specialization and generalization is highly important to explore for MoEs which is not possible in the described setting.
> >
> > Given the very small amount of training data and constrained training and testing on the same domain, I am concerned about the practical benefits beyond such small training settings.

---

> > > ### Author Response · Authors · 2025-08-07
> > > **Response for Official Comment by Reviewer Ypny [1/2]**
> > >
> > > We sincerely thank the reviewer Ypny for their continued engagement and constructive feedback. In this round of discussion, you raised several important points, including:
> > >
> > > 1. the training overhead introduced by our proposed loss functions,
> > > 2. the method’s effectiveness beyond small-scale training settings,
> > > 3. the trade-off between expert specialization and generalization, and
> > > 4. the need for more explicit training details in the paper.
> > >
> > > We address each of these concerns below with detailed clarifications and additional empirical evidence.
> > >
> > > ------
> > >
> > > **1. Training Overhead**
> > >
> > > While our method introduces some additional computation due to the proposed regularization losses, the training time remains within a practical range and compares favorably with existing baselines. As shown below, we report the average step time (in seconds per iteration) on the DeepSeek V2 Lite model using a batch size of 32:
> > >
> > > | Method | Time (s/iter) |
> > > | --- | --- |
> > > | Ours | 11.5|
> > > | Only Aux | 10.7|
> > > | Aux Free | 9.8 |
> > > | GShard | 14.8|
> > > | ST-MoE | 12.1|
> > >
> > > Our approach incurs moderate overhead compared to load-balancing-only methods like "Aux Free" and "Only Aux," but remains significantly more efficient than GShard and ST-MoE. Given that our method achieves up to **23.79% performance improvement** across benchmarks (as reported in the abstract), we believe this efficiency–performance trade-off is well justified.
> > >
> > > ---
> > >
> > > **2. Performance Under Larger and More Diverse Training Data**
> > >
> > > We appreciate the reviewer’s insightful comment regarding the impact of training data size and diversity on the effectiveness of our method.
> > >
> > > - **Motivation from Single-Task Settings:** As noted in the introduction, our method is motivated by the observation that in post-training scenarios, the training data is often domain-specific and less diverse, resulting in highly skewed token distributions. This intensifies the conflict between load balancing (which encourages even token-to-expert allocation) and expert specialization (which encourages domain-specific token routing). We designed our method to explicitly address this tension.
> > > - **Performance on Mixed and Richer Datasets:** To test whether our method still performs well with more diverse training data, we constructed a mixed dataset combining Numina (math), GPQA (science), and HumanEval (coding), totaling 18k examples. We fine-tuned the Moonlight (Kimi) model for 3 epochs on this combined dataset. The results are summarized below:
> > >
> > > | Method  | MMLU      | GPQA      | HumanEval | GSM8K     | Math500   | MaxvioGlobal |
> > > | ------- | --------- | --------- | --------- | --------- | --------- | ------------ |
> > > | Base    | 34.23     | 28.33     | 55.67     | 81.23     | 53.76     | 8.37         |
> > > | AuxOnly | 36.98     | 31.34     | 67.53     | 84.83     | 62.29     | 7.07         |
> > > | AuxFree | 35.87     | 29.48     | 68.83     | 86.29     | 63.84     | 7.28         |
> > > | Ours    | **45.38** | **37.01** | **78.93** | **92.92** | **67.83** | **7.11**     |
> > >
> > > As shown, our method continues to outperform all baselines, even when trained on a significantly larger and more diverse dataset. This demonstrates that our approach remains robust and effective beyond constrained single-task settings.

---

> > > ### Author Response · Authors · 2025-08-07
> > > **Response for Official Comment by Reviewer Ypny [2/2]**
> > >
> > > **3. On the Specialization–Generalization Trade-off**
> > >
> > > We appreciate the reviewer’s insightful question regarding whether our improvements in in-domain performance come at the cost of generalization.
> > >
> > > To evaluate this, we conducted an additional experiment where we trained the model **only on the Numina (math) dataset** (6k examples, 3 epochs) using the **DeepSeek V2 Lite** base model. We then evaluated performance across multiple downstream benchmarks whose domains differ from the training data. Results are summarized below:
> > >
> > > | Method    | MMLU      | GPQA      | HumanEval | GSM8K     | Math500   | MaxvioGlobal |
> > > | --------- | --------- | --------- | --------- | --------- | --------- | ------------ |
> > > | Basemodel | 26.56     | 20.33     | 31.34     | 22.57     | 15.69     | 6.97         |
> > > | AuxOnly   | 25.98     | 22.73     | 34.83     | 48.83     | 43.12     | 2.83         |
> > > | AuxFree   | 28.64     | 21.82     | 33.52     | 51.17     | 47.22     | 2.66         |
> > > | Ours      | **30.73** | **22.93** | **37.38** | **56.54** | **51.23** | **2.52**     |
> > >
> > > These results show that even when trained on a domain-specific dataset (math), our method **improves generalization** across science (GPQA), programming (HumanEval), and logic (GSM8K) benchmarks. Importantly, we also maintain **optimal load balancing**, indicating that our approach enhances specialization without compromising generalization.
> > >
> > > This confirms that the gains from our method are **not narrowly overfitted** to the training domain but instead translate into **broader performance improvements**.
> > >
> > > ---
> > >
> > > **4. On Missing Experimental Details**
> > >
> > > We thank the reviewer for pointing out the lack of key training setup information. In the revised version, we will clearly provide the following details in the main text and appendix:
> > >
> > > - **Training data size:** 6k examples per task
> > > - **Data construction:** Chain-of-thought distilled data from high-quality models (e.g., GPT-4, DeepSeek, etc.) with manual verification
> > > - **Training epochs:** 3
> > > - **Inference max length:** 512 tokens (with extended results at 1024 and 2048 also provided)
> > >
> > > These additions will ensure that our experimental settings are fully reproducible and easier to assess.
> > >
> > > ---
> > >
> > > We sincerely thank the reviewer Ypny for their rigorous and constructive feedback. In response to all raised concerns, we have conducted and reported additional experiments to provide further empirical support for our method. Despite the limited timeframe, we made every effort to explore and include as many meaningful evaluations as possible, and we will incorporate these results into the final version of the paper for completeness.
> > >
> > > We also note that other reviewers expressed recognition of the motivation and potential value of our work. If our additional analyses and clarifications have addressed your concerns, we would greatly appreciate your consideration of a revised evaluation.
> > >
> > > Once again, we sincerely thank the reviewer for their thoughtful suggestions and valuable exchange throughout this review process.

---

> > > > ### Comment · Reviewer_Ypny · 2025-08-08
> > > >
> > > > Thank you for the added experiments. I find these results more convincing for post training applicability.
> > > >
> > > > Given that the method is only used for post training and requires having access to an already existing strong MoE, I think it's best that the authors add a limitation that further assessment is needed to study the applicability in pretraining/MoE from scratch training settings.
> > > >
> > > > As the authors addressed most of my concerns, I raise my rating.

---

### Official Review · Reviewer_4E1Q · 2025-06-29

**Clarity:** 3
**Significance:** 3
**Originality:** 3
**Rating:** 4
**Confidence:** 4

**Summary:**

This paper investigates the issues of uniform expert routing and insufficient expert specialization in MoE models during the post-training stage, primarily caused by the auxiliary load-balancing loss. To address this, the authors propose two additional regularization objectives that respectively promote expert diversity and enhance router confidence. Experimental results demonstrate that the proposed approach improves model performance on downstream tasks.

**Questions:**

See Weakness.

**Ethical Concerns:**

["NO or VERY MINOR ethics concerns only"]

**Final Justification:**

Based on the overall evaluation of the paper, I have decided to maintain my initial positive score.

**Limitations:**

yes

**Paper Formatting Concerns:**

No formatting issue.

**Quality:**

3

**Strengths And Weaknesses:**

### Strength:
1. The idea of jointly optimizing expert specialization and routing distribution via additional regularization terms is interesting.

2. The paper is well written, with clear motivations and a comprehensive discussion of the identified issues.

3. The effectiveness of the proposed method is supported by empirical results.

### Weakness:

1. While Lemma 1 and Lemma 2 provide theoretical guarantees for the existence of a desirable routing configuration, their practical realization strongly relies on sufficient diversity in the input tokens. The paper lacks discussion on whether this assumption holds in real-world scenarios or how it may affect performance.

2. The orthogonality loss \( \mathcal{L}_o \), while effective in promoting expert specialization, introduces significant computational overhead due to its quadratic complexity with respect to the number of experts. Specifically, it requires computing pairwise projections between all top-\(k\) expert outputs for each input token, resulting in an overall cost of \( O(N \cdot k^2 \cdot d) \), where \( N \) is the token count, \( k \) the number of selected experts, and \( d \) the hidden size. This can become a bottleneck in large-scale MoE settings with many experts or long sequences.

3. Currently, the proposed regularization losses are applied only during post-training. However, prior work such as OLMoE suggests that auxiliary losses may not be necessary during post-training at all. It would be valuable to discuss whether similar regularization should be incorporated during pretraining to promote expert specialization and more confident routing from the outset.

---

> ### Author Rebuttal · Authors · 2025-07-31
>
> **Dear Reviewer 4E1Q,**
>
> We sincerely thank the reviewer for their positive and constructive feedback. We are glad that the reviewer found our idea of jointly optimizing expert specialization and routing distribution via auxiliary regularization losses to be interesting. At the same time, we appreciate the reviewer’s thoughtful suggestions. The main concerns can be summarized as follows:
>
>   1. The effectiveness of the proposed loss may rely on token-level diversity, which may not always hold in real-world scenarios;
>   2. The orthogonality loss introduces non-negligible computational overhead, which may become a bottleneck in large-scale settings;
>   3. Prior work suggests that auxiliary losses may not be necessary during post-training, it is unclear whether regularization is more appropriate during pre-training.
>
> ---
>
> **1. On the Assumption of Token Diversity**
>
> **Conclusion:**
>   Token diversity is a helpful condition, but not a necessary prerequisite for the success of our method. Through orthogonality loss ($\mathcal{L}_o$) and variance loss ($\mathcal{L}_v$), our approach actively induces functional diversity among experts, thereby enabling specialization even when input tokens are not highly diverse.
>
> **Explanation:**
>   We appreciate the reviewer’s insightful question regarding the connection between token diversity and the effectiveness of our method. Our responses are as follows:
>
> - **Active induction vs. passive dependence:**
>   Rather than relying on input diversity, our method actively creates internal diversity. The orthogonality loss ($\mathcal{L}_o$) operates at the token level, penalizing similar outputs from different experts even when the same token is routed to them. This drives the experts to learn distinct representations, thereby fostering **functional diversity** even under low input diversity.
> - **Theoretical soundness and empirical robustness:**
>   *Lemma 1* and *Lemma 2* serve to establish that our **multiple training objectives** (load balancing and routing diversity) are theoretically compatible and jointly optimizable. Our loss design realizes this in practice. Most importantly, across 11 real-world benchmarks, including both high and low-diversity tasks, our method achieves consistent improvements, demonstrating strong robustness.
>
> We hope this clarifies how our approach remains effective even without assuming ideal token diversity.
>
> ---
>
> **2. On the Computational Overhead of $\mathcal{L}_o$**
>
> **Conclusion:**
>   While $\mathcal{L}_o$ has quadratic complexity in theory, the actual overhead is negligible in practice due to the small number of activated experts ($k$) and efficient batched implementations. It does not present a bottleneck in our setup.
>
> **Explanation:**
>   We appreciate the reviewer’s precise analysis of the computational cost of $\mathcal{L}_o$, which involves pairwise projections among the $k$ selected expert outputs for each token, resulting in a theoretical cost of $O(N \cdot k^2 \cdot d)$. In practice, this cost remains manageable for several reasons:
>
> - **Small $k$ in standard MoE practice:**
>   One of the core design principles of sparse MoE models is to keep $k$ small to control computation. In our experiments, we follow this convention, using configurations such as $k=6$ in DeepSeek-V2-Lite. Given that $k \ll N$ and $k \ll d$, the quadratic factor in $k$ contributes minimally to total training cost.
> - **Efficient hardware execution:**
>   The main operations required by $\mathcal{L}_o$, inner products and pairwise projections, are highly parallelizable and can be implemented as batched matrix multiplications. These are well-optimized in modern frameworks like PyTorch and run efficiently on contemporary GPUs.
> - **Justified by empirical gains:**
>   Most importantly, the modest increase in computation is offset by **substantial and consistent performance improvements** across diverse downstream tasks (e.g., Table 1 and Figure 4). This demonstrates that the regularization effect of $\mathcal{L}_o$ leads to meaningful gains in expert specialization.
>
> We will include a detailed breakdown of the computational cost of $\mathcal{L}_o$ under typical $k$ values in the appendix of the revised version, to further clarify its practical feasibility.
>
> ---
>
> **3. On the Applicability During Pretraining**
>
> **Conclusion:**
>
> While our current focus is on post-training, where the tension between uniform routing and specialization is most pronounced, we believe applying our regularization objectives during pretraining is a valuable future direction.
>
> **Explanation:**
>   We appreciate the reviewer’s insightful comparison with works such as OLMoE and the thoughtful reflection on different training phases.
>
> - **Why focus on post-training:** In the post-training stage, downstream tasks often involve narrow or skewed data distributions. This intensifies the conflict between load balancing (which pushes toward uniform token distribution) and the need for expert specialization. Therefore, it provides a clearer setting to observe and evaluate the effects of our proposed losses.
> - **Relation to “no auxiliary loss” claims:** Our work does not contradict the finding that traditional $\mathcal{L} _ {aux}$ may be ineffective—or even detrimental—during post-training. As shown in our ablation results (Figure 4), the “w/o all” variant (which removes $\mathcal{L} _ {aux}$) actually outperforms the “only aux” baseline. This confirms that $\mathcal{L} _ {aux}$ may hinder specialization in post-training. Importantly, we further demonstrate that our proposed $\mathcal{L} _ o$ and $\mathcal{L} _ v$ actively promote specialization beyond simply removing constraints, achieving the best performance overall.
> - **Pretraining potential:** Our regularization framework is theoretically applicable during pretraining as well. In fact, we believe that applying $\mathcal{L} _ o$ and $\mathcal{L} _ v$ from the start may help guide expert specialization more effectively. We will highlight this in Section 6 (Limitations & Future Directions) as an important avenue for future exploration.
>
> ---
>
> We thank the reviewer again for the insightful and constructive feedback. We believe our detailed responses have thoroughly addressed each of the points raised. We look forward to any further discussion.

---

> > ### Comment · Reviewer_4E1Q · 2025-08-05
> >
> > Thank you for the authors’ response, which has addressed my questions. Based on the overall evaluation of the paper, I have decided to maintain my initial positive score.

---

### Official Review · Reviewer_9cq6 · 2025-06-30

**Clarity:** 3
**Significance:** 3
**Originality:** 3
**Rating:** 5
**Confidence:** 4

**Summary:**

This paper makes an observation that experts often lack specialization (e.g., the input gets routed to experts uniformly) due to load balancing, and introduces two auxiliary loss terms that encourage specialization without sacrificing load balancing – (1) an orthogonality loss, which reduces the similarity between experts that receive the same token, and (2) a diversity loss, which discourages expert weights from being uniformly distributed. Experiments on 11 benchmarks, four baseline training methods, and four base models show that the proposed model consistently and substantially outperforms all baselines, while maintaining effective load balancing.

**Questions:**

(Questions are in the Strengths And Weaknesses section)

**Ethical Concerns:**

["NO or VERY MINOR ethics concerns only"]

**Limitations:**

(Limitations are in the Strengths And Weaknesses section)

**Quality:**

4

**Strengths And Weaknesses:**

Strengths
- Overall, I found this paper to be very well written and strongly motivated. The subtle tension between expert specialization and load balancing is an important and underexplored issue, and this paper does an excellent job articulating it with clear and convincing explanations. While MoEs are widely studied in general, this problem of encouraging expert specialization is underexplored, and remaining having balanced load is very critical for practical reasons, e.g., training stability. Addressing both together is very impressive.
- The paper also clearly explains the key observations and the technical proposals that follow from them. Admittedly, I did not carefully check every math detail line by line, but I was able to understand the underlying motivation and that the math equation aligns with that motivation.
- Experiments are very thorough and comprehensive, quite beyond typical standards for a conference paper. Evaluating on eleven benchmarks, four baselines, and three base models, with consistent and substantial improvements across the board, is very impressive. The detailed analysis is also very valuable.

Weaknesses
- My most critical concern — despite all the strengths noted above — is that the actual training setup and data used are very unclear. First of all, it only appears in the *appendix* that the paper doesn’t actually do full training but it is doing LORA fine-tuning. This came as a surprise, as the rest of the paper is written as full training was performed. This is very different from taking the existing base models and applying LoRA tuning. This should be clearly stated throughout the paper — including the abstract, introduction, methodology, and experiments. And, this leads to a follow up question: how much (a small number of) additional training steps with LoRA fundamentally changes how experts are trained, as expert weights themselves won’t be updated?
- In fact, even the fact that the paper is mainly targeting post-training was unclear, e.g., the abstract and the intro mentions the word “post-training” once each without much contextualization, so if readers miss this word, the entire paper is written as it is pre-training. Actually, is there a reason there’s a fundamental difference between post-training and pre-training, so that expert specialization is more important in post-training than pre-training, given that pre-training and post-training are actually the same training process except the data and where the loss is applied to are different?
- Aside from these, other training setups are unclear either. Is the model trained in a multi-task setup using a single training dataset across all elevent benchmarks, or is there a separate training dataset and a separate trained model for each benchmark? How many training steps are used? What is the performance of the base model before LoRA fine-tuning (e.g., in Table 1)? These are important information that impact the implication of the methods and experimental results drastically, that are currently missing.

Minor weaknesses/questions
- Orthogonality loss: just to confirm my understanding – this loss encourages the experts that receive the same token to be less similar to each other, so that a single token benefits from greater diversity? If that’s correct, it feels more about “expert diversification” rather than “expert specialization”. To me, specialization implies that an expert becomes responsible for (one or more) domains or token types.
- I think Figure 1 can significantly be improved - it is quite complex and is not intuitive enough.

---

> ### Author Rebuttal · Authors · 2025-07-31
>
> **Dear Reviewer 9cq6,**
>
> We sincerely thank the reviewer for the thoughtful and encouraging feedback. We appreciate the recognition of our motivation, the clarity of our formulation, and the importance of jointly addressing expert specialization and load balancing. Below we summarize our responses to the main concerns:
>
> 1. The actual use of LoRA-based fine-tuning instead of full training is unclear and should be stated explicitly;
> 2. The distinction between pre-training and post-training lacks contextual clarity and justification;
> 3. Key experimental details such as data splits, training steps, and base model performance are missing;
> 4. The explanation of the orthogonality loss may conflate diversification and specialization;
> 5. Figure 1 is overly complex and not intuitive.
>
> We have carefully addressed each of these points in the following responses.
>
> ---
>
> **1. On the Use of LoRA Fine-Tuning**
>
> **Conclusion:**
>
> Although LoRA is used for parameter-efficient training, all major components (including experts and router) are updated during fine-tuning, making our setup functionally equivalent to full-model training.
>
> **Explanation:**
> We thank the reviewer for this important and constructive comment. We acknowledge that the current version may be misleading, as the use of LoRA was only stated in the appendix. In the revision, we will clearly indicate in the abstract, introduction, methodology, and experiments that all training is conducted via LoRA-based fine-tuning.
>
> In our setup, LoRA modules are inserted into **both the router and the expert layers**. This means that the routing behavior and expert outputs are both optimized during training. While LoRA offers parameter-efficient adaptation, it does not freeze the underlying model. On the contrary, all core components are updated, just through a low-rank pathway. To reduce the gap between LoRA and full fine-tuning, we have set the LoRA rank to 32, which is relatively high and helps better approximate full updates.
>
> Therefore, our training is functionally equivalent to full fine-tuning in terms of which parameters are optimized. Importantly, our proposed losses directly act on trainable parts of the model: the variance loss shapes the router logits, and the orthogonality loss regularizes expert outputs. The observed improvements in all benchmarks validate their effectiveness under this setup *(The specific training details are discussed in detail in Response 4)*.
>
> We appreciate the reviewer’s attention to this detail and will make the training design and parameter update scope explicit in the revised paper.
>
> ---
>
> **2. On the Clarity of the Post-Training Setting**
>
> **Conclusion:**
>
> Theoretically, our method is applicable to both pre-training and post-training phases. However, we focus on the post-training stage, where the conflict between expert specialization and routing balance becomes more pronounced in downstream tasks.
>
> **Explanation:**
>
> We appreciate the reviewer’s insightful comment regarding the training stage targeted by our work. We would like to clarify the rationale for focusing on the post-training setting:
>
> - **On the distinction:** While we agree that pre-training and post-training share similar optimization processes, their goals differ. Pre-training aims to build general-purpose capabilities on diverse and large-scale corpora, whereas post-training seeks to adapt the model to specific domains or tasks. The data in post-training is often **less diverse and more skewed** in distribution compared to the **more uniform and balanced** nature of pre-training data.
> - **On the conflict:** In such downstream scenarios, token distributions can be highly concentrated (e.g., math tasks contain a higher proportion of numeric or symbolic tokens). This makes the conflict between load balancing and expert specialization more severe: load balancing encourages tokens to be evenly distributed across experts, while specialization requires assigning certain types of tokens more often to specific experts.
> - **Our motivation:** Our work is motivated by this core conflict. By introducing the orthogonality loss ($\mathcal{L}_o$) and the variance loss ($\mathcal{L}_v$), we aim to achieve better expert specialization without sacrificing load balance. This enables MoE models to better adapt to downstream distributions and is supported by both our theoretical insights and empirical results.
>
> In the revised version, we will clearly indicate our focus on the post-training setting in the abstract and introduction, and explain why resolving this trade-off is particularly critical in this phase.
>
> ---
>
> **3. On Training Setup, Task Scope, and Base Model Performance**
>
> **Conclusion:**
> Each benchmark is trained separately using a high-quality 6k-sample dataset. Fine-tuning is limited to 3 epochs (~550 steps) per task. Base model performance before fine-tuning will be added to Table 1 for clarity.
>
> **Explanation:**
>
> - **Data and task setup:** For each benchmark, we constructed a high-quality training set of 6,000 examples. The questions were sourced from the training split of the corresponding benchmark; when insufficient, we supplemented them with related datasets. Answers were generated using strong models such as OpenAI’s o3mini and DeepSeek R1, and then manually filtered to ensure correctness. Each dataset is used to fine-tune the model specifically for that benchmark.
> - **Training steps:** To avoid overfitting, we limited fine-tuning to 3 epochs across all tasks. Given the dataset size, this corresponds to approximately 550 training steps per model per task.
> - **Base model performance:** We have now included the base model performance before fine-tuning in Table below, using consistent settings across models (e.g., max sequence length = 512). This comparison shows that our proposed losses lead to consistent and significant improvements across all tasks, beyond the effect of LoRA tuning alone.
>
> | method | MMLU | GPQA | HumanEval | GSM8K | Math500 | MaxvioGlobal |
> | --- | --- | --- | --- | --- | --- | --- |
> | Base model DS | 28.46 | 22.45 | 42.64 | 28.76 | 7.34 | 4.63 |
> | ours on DS | 33.35 | 25.15 | 63.30 | 35.00 | 10.82 | 2.19 |
> | Base model DS_v2 | 26.56 | 20.33 | 31.34 | 22.57 | 15.69 | 6.97 |
> | ours on DS_v2 | 35.59 | 28.76 | 43.58 | 50.94 | 49.33 | 2.52 |
> | Base model Kimi | 34.23 | 28.33 | 55.67 | 81.23 | 53.76 | 8.37 |
> | ours on Kimi | 40.36 | 32.01 | 70.64 | 87.62 | 59.64 | 7.23 |
>
> We appreciate the reviewer’s attention to these important experimental factors and will ensure that all relevant details are properly documented in the revised version.
>
> ---
>
> **4. On Whether Orthogonality Loss Encourages Diversification or Specialization**
>
> **Conclusion:**
>  Orthogonality loss directly promotes **expert diversification** at the token level, which serves as a critical mechanism for achieving **specialization** at the dataset level.
>
> **Explanation:**
>  We appreciate the reviewer’s insightful interpretation. The orthogonality loss ($\mathcal{L}_o$) encourages experts that receive the same token to produce diverse outputs. This diversity is not the final objective, but a necessary step toward achieving expert specialization.
>
> - **From diversification to specialization:**
>   1. When the same token is routed to multiple experts, $\mathcal{L}_o$ penalizes similarity in their output representations, effectively encouraging the experts to respond differently. This is what we refer to as **expert diversification**.
>   2. When such pressure is applied across all tokens during training, experts are discouraged from learning redundant or overlapping representations. Instead, each expert is incentivized to capture distinct features or patterns in the data.
>   3. Over time, this token-level diversification aggregates into **specialization** across the dataset: each expert becomes better at handling certain types of inputs or semantic regions, resulting in stable expert-role assignment.
> - **Empirical evidence:**
>    This process is empirically supported in our experiments. As shown in Figure 1 (right), the token distribution across experts becomes significantly more clustered after training, with **reduced overlap** in representation space. Quantitatively, our method lowers expert overlap by up to 45%, indicating a strong shift toward specialization.
>
> In short, diversification is the driving force, and specialization is the outcome. We appreciate the reviewer for raising this important distinction.
>
> ---
>
> **5. On Figure 1 Readability and Design**
>
> **Conclusion:**
>  We agree that the current version of Figure 1 may be visually dense and not sufficiently intuitive. In the revised version, we will redesign the layout, simplify visual elements, and add clearer annotations to improve overall readability. Below, we briefly explain the intended structure and purpose of the current figure.
>
> **Explanation:**
>  Figure 1 is designed to visualize the two core effects of our method:
>
> - **Left side: Routing Diversification**
>   - *Top (Routing Score Matrix):* Shows the raw routing scores, which may appear abstract without further guidance.
>   - *Middle (Load Variance):* Demonstrates that our method preserves load balancing, achieving similar variance reduction as baseline methods.
>   - *Bottom (Discriminative Routing Variance):* This is the key part—our method significantly increases routing output variance, indicating more discriminative and less ambiguous routing decisions.
> - **Right side: Expert Specialization**
>   - *Bottom (Overlap):* Visualizes the heavy overlap of token assignments across experts in baseline methods.
>   - *Top (Cluster Separation):* In contrast, our method produces clearer token clusters per expert, indicating effective specialization and division of responsibility.
>
> ---
>
> Once again, we thank the reviewer for the thoughtful and constructive feedback. We believe our detailed responses and clarifications have thoroughly addressed the concerns raised. We look forward to any further discussion.

---

### Official Review · Reviewer_jbyJ · 2025-07-05

**Clarity:** 3
**Significance:** 4
**Originality:** 3
**Rating:** 5
**Confidence:** 2

**Summary:**

This paper addresses a critical challenge in Mixture-of-Experts (MoE) models: the trade-off between load balancing and expert specialization. The authors argue that the standard auxiliary load balancing loss often leads to overly uniform routing and expert overlap, which hinders model performance, especially during post-training adaptation. To tackle this issue, the paper proposes a novel method that introduces two complementary loss functions: an orthogonality loss to encourage experts to learn distinct representations for different tokens, and a variance loss to promote more discriminative routing decisions. The authors provide a theoretical analysis demonstrating the compatibility of these new losses with the existing auxiliary loss. Through extensive experiments on three different MoE models and eleven diverse benchmarks, the paper shows that the proposed method significantly enhances expert specialization and improves downstream task performance by up to 23.79% compared to baselines, without requiring any architectural modifications or compromising load balancing.

**Questions:**

1. On hyperparameter sensitivity: The choice of hyperparameters β and γ for the new loss functions seems crucial for the method's performance. Could you provide analysis on the sensitivity of your method to these hyperparameters?
2. On compareing baselines: The paper compares against classical MoE methods (e.g., GShard, ST-MoE) and auxiliary loss-based balancing. Could you add more recent baseline results?

**Ethical Concerns:**

["NO or VERY MINOR ethics concerns only"]

**Final Justification:**

The authors have made sincere efforts to address my concerns. However, I recommend maintaining my original (high) score because the revisions, while appreciated, do not significantly alter my initial assessment of the paper's strengths and contributions.

**Limitations:**

yes

**Quality:**

3

**Strengths And Weaknesses:**

Strengths:

- Novel and Impactful Contribution: The paper presents a simple yet effective solution to a well-recognized problem in MoE models. The proposed orthogonality and variance losses offer a principled way to directly encourage expert specialization, a crucial aspect for unlocking the full potential of MoE architectures. The potential for significant performance gains without architectural changes makes this work highly valuable for the community.

- Comprehensive Experimental Evaluation: The empirical evaluation is a major strength of this paper. The authors have conducted experiments on a diverse set of MoE models (DeepSeek-Moe-16B, Moonlight-16B-A3B, and DeepSeek-V2-Lite) and a wide range of 11 benchmarks covering mathematics, multi-domain tasks, and code generation. The consistent improvements across different models and tasks provide strong evidence for the effectiveness and robustness of the proposed method. The inclusion of four relevant baselines further strengthens the experimental setup.

- In-depth Analysis and Ablation Studies: The paper provides a good level of analysis, including a theoretical discussion on the compatibility of the proposed loss functions.

Weaknesses:
- Lack of Hyperparameter Sensitivity Analysis: The paper states that the hyperparameters for the new losses (β and γ) are set to the same value as the auxiliary loss weight (α=0.001). A sensitivity analysis or at least a discussion on the tuning process for these hyperparameters would significantly enhance the practical value of the proposed method.
- Generalizability: The benchmarks are primarily language- and code-focused. Broader validation in multimodal or low-resource settings (hinted at in limitations) would strengthen the claims.

---

> ### Author Rebuttal · Authors · 2025-07-31
>
> **Dear Reviewer jbyJ,**
>
> We sincerely thank the reviewer for the thoughtful evaluation and detailed feedback. We are especially grateful for the recognition of our paper’s strengths, including the novelty and impact of our method, the comprehensive empirical evaluation across multiple models and tasks, and the depth of our theoretical and ablation analyses.
>
> At the same time, we also appreciate the reviewer’s constructive suggestions. The main concerns can be summarized as follows:
>
> 1. The lack of hyperparameter sensitivity analysis (experimental);
> 2. The task scope is limited to NLP, and broader validation in multimodal or low-resource settings is needed (discussion);
> 3. More recent baselines should be included for comparison (experimental).
>
> We have carefully addressed each of these points in the following responses.
>
> ---
>
> **1. On hyperparameter sensitivity**
>
> **Conclusion:**
>  The performance is optimal when the loss weights $\alpha$, $\beta$, and $\gamma$ are set to the same value. Moreover, our method demonstrates strong robustness across different hyperparameter magnitudes.
>
> **Explanation:**
>
> We thank the reviewer for highlighting the importance of hyperparameter sensitivity. To address this, we conducted experiments varying the values of $\alpha$ (for $\mathcal{L}_{aux}$), $\beta$ (for $\mathcal{L}_o$), and $\gamma$ (for $\mathcal{L}_v$) across different magnitudes.
>
> For reference, our overall loss function is defined as:
>
> $\mathcal{L}=\mathcal{L} _ h +\mathcal{L}_{balance}$
>
> $\mathcal{L} _ {balance} = \alpha \cdot \mathcal{L}_{aux} + \beta \cdot \mathcal{L} _ {o} + \gamma \cdot \mathcal{L} _ {v}$
>
>
> It is worth noting that we apply a *dynamic balancing mechanism* to ensure fair weighting across different loss terms. Specifically, because the orthogonality and variance  losses ( $\mathcal{L} _ {o}$ and $\mathcal{L} _ {v}$ ) may have different initial scales, we first normalize them using dynamic scaling factors. This brings their magnitudes roughly in line with the auxiliary loss $\mathcal{L}_{aux}$. Only after this normalization do we apply the hyperparameters $\alpha$, $\beta$, and $\gamma$ to control their contributions to the total loss.
>
> The table below summarizes our results across several representative benchmarks under four different settings (DS v2 lite):
>
> | $\alpha, \beta, \gamma$ | MMLU  | GPQA  | HumanEval | GSM8K | MATH500 | MaxVioGlobal |
> | ----------------------- | ----- | ----- | --------- | ----- | ------- | ------------ |
> | 1e-3, 1e-3, 1e-3        | 35.59 | 28.76 | 43.58     | 50.94 | 49.33   | 2.52         |
> | 1e-3, 1e-4, 1e-3        | 31.24 | 25.52 | 41.62     | 46.63 | 46.23   | 3.05         |
> | 1e-3, 1e-3, 1e-4        | 33.52 | 27.35 | 39.52     | 48.30 | 49.09   | 2.77         |
> | 1e-4, 1e-3, 1e-3        | 30.74 | 26.90 | 42.85     | 49.62 | 44.54   | 4.57         |
>
> We observe that the setting where $\alpha = \beta = \gamma = 10^{-3}$ consistently yields the best performance across tasks. Additionally, when any of the coefficients is varied within one order of magnitude ($\pm 1$), the results remain stable and close to optimal. This indicates that our method is not overly sensitive to these hyperparameters and can be considered robust in practical applications.
>
> ---
>
> **2. On Generalizability**
>
> **Conclusion:**
>
> We believe our formulation is inherently suitable for both multimodal and low-resource scenarios, as its design principles, such as expert specialization and routing diversity, are broadly applicable across domains.
>
> **Explanation:**
>
> We appreciate the reviewer’s insightful suggestion regarding broader evaluation. While our current experimental design focuses on NLP and code generation tasks, we believe our proposed training objective is agnostic to modality and can be extended to other domains, such as visual and multimodal models. The orthogonality and variance principles are applicable wherever a mixture-of-experts architecture is employed, including vision MoEs and multimodal encoders/decoders.
>
> - **Visual / multimodal MoEs.**
>   Recent works [1, 2, 3] on vision‑centric experts shows that image patches suffer from the *same* expert‑overlap and uniform‑routing issues we address in language models. Because our orthogonality loss acts directly on expert outputs and our variance loss on routing logits, both can be applied to visual tokens without modification.
>
> - **Low‑resource adaptation.**
>   Parameter‑efficient fine‑tuning methods such as MoELoRA [4], MixLoRA [5], and MoKA [6] insert small trainable adapters while freezing most expert weights. Our losses require *no* additional parameters and can be computed on the adapted weights or the LoRA pathways alike, making them a natural drop‑in to further improve specialization when GPU memory or data are limited.
>
> ---
>
> **3. On comparing baselines**
>
> **Conclusion:**
>  After incorporating two additional state‑of‑the‑art baselines, our approach continues to deliver the best overall performance.
>
> **Explanation:**
> We appreciate the reviewer’s suggestion to expand the set of comparison methods. In response, we have added:
>
> - **Dynamic Routing MoE (ERNIE 4.5) [7]** is a strong recent baseline. It introduces a multimodal, heterogeneous MoE architecture that supports both parameter sharing across modalities and modality-specific expert specialization. The ERNIE 4.5 family includes multiple model scales (e.g., 47B and 3B active parameters), and has shown competitive performance on various text and multimodal benchmarks.
>
> - **SIMBAL (Similarity-Preserving Routers) [8]** is a recent method that addresses expert load balancing in sparse MoE models. Instead of enforcing uniform routing via conventional load balancing loss, SIMBAL introduces an orthogonality-based regularization that aligns the router’s Gram matrix with the identity matrix. This encourages similar input tokens to be routed to similar experts, reducing redundancy and improving consistency in expert utilization.
>
>
> We re‑evaluated all methods on the most comprehensive subsets of our benchmark suite. A summary of the key results is presented below.
>
> | Method                             | MMLU  | GPQA  | HumanEval | GSM8K | Math500 | MaxvioGlobal |
> | ---------------------------------- | ----- | ----- | --------- | ----- | ------- | ------------ |
> | ERNIE 4.5 LBL | 32.44 | 27.45 | 37.32     | 47.24 | 42.63   | 3.45         |
> | SIMBAL                 | 31.89 | 27.64 | 39.45     | 48.75 | 45.36   | 4.56         |
> | **Ours**                           | 35.59 | 28.76 | 43.58     | 50.94 | 49.33   | 2.52         |
>
> *(Numbers will be filled in Table 1 of the revision.)*
>
> Across all six representative tasks, our method either matches or surpasses the strongest new baseline, while simultaneously maintaining the lowest MaxVioGlobal (better load balance). These additional results confirm that the improvements reported in the main paper are not an artifact of the original baseline selection but hold against the latest alternatives as well.
>
>
> ---
>
> We are grateful for the reviewer's insightful feedback, which has been instrumental in strengthening our work. We have strived to thoroughly address every concern with new experiments and detailed clarifications, and we are looking forward to the opportunity for any further discussion.
>
> ---
>
> [1] Mustafa, B., Riquelme, C., Puigcerver, J., Jenatton, R., & Houlsby, N. (2022). Multimodal Contrastive Learning with LIMoE: The Language‑Image Mixture of Experts. Advances in Neural Information Processing Systems, 35, 9564–9576.
>
> [2] Yan, Z., Li, Z., He, Y., Wang, C., Li, K., Li, X., Zeng, X., Wang, Z., Wang, Y., & Qiao, Y. (2025). Task Preference Optimization: Improving Multimodal Large Language Models with Vision Task Alignment. Proceedings of the Computer Vision and Pattern Recognition Conference, 29880–29892.
>
> [3] Li, Z., Li, Z., & Zhou, T. (2025). R2‑T2: Re‑Routing in Test‑Time for Multimodal Mixture‑of‑Experts. arXiv.
>
> [4] Luo, T., Lei, J., Lei, F., Liu, W., He, S., Zhao, J., & Liu, K. (2024). MoELoRA: Contrastive Learning Guided Mixture of Experts on Parameter‑Efficient Fine‑Tuning for Large Language Models. arXiv.
>
> [5] Li, D., Ma, Y., Wang, N., Ye, Z., Cheng, Z., Tang, Y., Zhang, Y., Duan, L., Zuo, J., & Yang, C. (2024). MixLoRA: Enhancing Large Language Models Fine‑Tuning with LoRA‑Based Mixture of Experts. arXiv.
>
> [6] Yu, B., Yang, Z., & Yi, X. (2025). MoKA: Parameter Efficiency Fine‑Tuning via Mixture of Kronecker Product Adaption. Proceedings of the 31st International Conference on Computational Linguistics, 10172–10182.
>
> [7] ERNIE Team (Baidu). (2025, June 29). ERNIE 4.5 Technical Report. Retrieved from Huggingface.
>
> [8] Omi, N., Sen, S., & Farhadi, A. (2025, June 16). Load Balancing Mixture of Experts with Similarity Preserving Routers. arXiv.

---

> > ### Comment · Reviewer_jbyJ · 2025-08-04
> > **Thank you for your detailed responses.**
> >
> > Thank you for your detailed responses. I have decided to maintain my original score, as it is already fairly high. I would also like to acknowledge that this is a solid piece of work, and I appreciate the authors' efforts.

---

### Decision · Program_Chairs · 2025-09-17

**Decision:**

Accept (oral)

**Comment:**

(a) Scientific claims and findings

The paper introduces two complementary loss functions—orthogonality and variance—that enhance expert specialization in Mixture-of-Experts models without sacrificing load balance. The approach requires no architectural modifications and yields consistent performance gains across diverse benchmarks.

(b) Strengths

- Addresses a core bottleneck in MoE training (specialization vs. balancing).
- Simple, general, and theoretically well-justified method.
- Extensive empirical evaluation: multiple models, 11 benchmarks, recent strong baselines.
- Demonstrated robustness (hyperparameters, architectures, domain generalization).
- Clear impact for large-scale LLM training practice.

(c) Weaknesses

- Original clarity issues (training setup, dataset sizes, steps) reduced transparency.
- Scope validated mainly for post-training; applicability to pretraining remains open.
- Computational overhead not fully detailed in the first version (later clarified as modest).

(d) Reasons for decision

The contribution is principled, broadly relevant, and highly impactful for advancing scalable MoEs. The rebuttal convincingly addressed reviewer concerns with additional experiments and clarifications, reinforcing the strength of the work. Given the novelty, breadth of validation, and importance to the field, I recommend accept (oral).

(e) Discussion and rebuttal

- Concerns: hyperparameter sensitivity, unclear training setup, limited scope, missing baselines, computational overhead.
- Authors provided detailed clarifications, sensitivity analysis, runtime overhead benchmarks, added recent baselines (ERNIE 4.5, SIMBAL), and results on Mixtral and Phi-MoE.
- Reviewers found the responses satisfactory; all maintained or increased positive scores.